# COHESIV: Contrastive Object and Hand Embeddings for Segmentation In Video

**Dandan Shan**[*]
University of Michigan
dandans@umich.edu

**Richard E.L. Higgins**[*]
University of Michigan
relh@umich.edu

**David F. Fouhey**
University of Michigan
fouhey@umich.edu

## Abstract

In this paper we learn to segment hands and hand-held objects from motion. Our system takes a single RGB image and hand location as input to segment the hand and hand-held object. For learning, we generate responsibility maps that show how well a hand's motion explains other pixels' motion in video. We use these responsibility maps as pseudo-labels to train a weakly-supervised neural network using an attention-based similarity loss and contrastive loss. Our system outperforms alternate methods, achieving good performance on the 100DOH, EPIC-KITCHENS, and HO3D datasets.

## 1 Introduction

We invite you to pick up an object in your vicinity and bring it towards you. As you hold the object and move your hand, the object moves coherently, and is roughly rigidly attached to a coordinate frame in your hand. While your adult brain does not need this signal and can readily differentiate the object in your hand from both your hand and the background even if you hold your hand still, how did you learn to do this? One answer [42] is that as a human you have time locked modalities of your own hand's configuration via proprioception and vision, and your hand and the object share a "common fate" [47]. The goal of this paper is to operationalize this idea by learning to segment hand and in-hand objects (irrespective of name) from a single image by learning from video data.

This goal poses many challenges for current computer vision. While there has been a longstanding interest in learning segmentation and object individuation from motion cues in vision [41, 34, 11], the general case of being broadly able to segment everything has not made substantial progress. While there has, of course, been substantial progress on instance segmentation for particular categories (for instance MaskRCNN [21]) powered by large segmentation datasets [28], the space of categories that one can pick up is vast. Indeed, recent work that aims to reconstruct in-hand objects via segmentation losses [7] reconstructs a few objects by finding category correspondences (e.g., tennis racket masks are used for knives). At the same time, while there is work on understanding hands, including contact state [32], and detecting boxes around held generic objects [40], there is no work on *segmenting* objects, much less work that learns from video data. In sum – segmenting lots of generic objects is still challenging, and segmenting generic objects in contact is no exception.

Our approach makes progress on this challenging problem by assuming a small amount of knowledge about humans. We assume we can estimate landmarks on hands [37], identify whether they are holding something [40], as well as identify which pixels belong to people [23]. This small amount of information guides understanding of the the deluge of information from optical flow [43]. Rather than segmenting the full 3D motions of 3D objects from the flow, we instead only have to identify whether the flow is *better* explained by a moving hand or a background. Moreover, by knowing about an ubiquitous object (the hand), we show we can learn about many less ubiquitous objects.

---

[*]Equal contribution.

35th Conference on Neural Information Processing Systems (NeurIPS 2021).

We implement our solution with a network, named *COHESIV*, that takes an RGB image and one 2D location on a hand as inputs to segment the hand and the object that the hand is interacting with (Section 3). At test time, *COHESIV* maps the input image to an embedding space with a CNN [38]; this embedding is processed with lightweight heads to produce per-pixel detectors at each hand, and feature maps for objects that can be queried to produce a heatmap. At training time, we use the motion with nearby frames to derive signal. We define a *responsibility map* for a moving hand as the relative goodness of fit on optical flow for a planar motion model in comparison to a background model. These responsibilities power two losses: a similarity loss that directly supervises per-hand segments; a contrastive loss [13] that encourages in-group affinity as well as separates embeddings among people, objects, and background via pseudo-labels.

We train and validate on video data of humans engaged in complex behaviors using subsets of the 100 Days of Hands (100DOH) [40], EPIC-KITCHENS-55 (EPICK) [9, 10], and HO3D [18] datasets. We compare with alternate methods that range from fully-supervised bounding boxes [40] to basic motion cues from optical flow [43], to saliency [51]. We show that our weakly-supervised method is comparable to the supervised bounding box detector method, while outperforming flow and saliency methods.

## 2   Related Work

Our work aims to segment hands and in-hand objects from a single image and thus interacts with a variety of related areas, ranging from: the domain we work on (understanding hand-object contact), the signals we use (common fate with optical flow), and the methods we use to extract meaning from this signal (contrastive learning). Our approach differs from each by using contrastive learning to extract supervision from optical flow in order to segment out objects people are holding.

Understanding hands in contact with objects has long been of interest. In addition to work focused on finding hands [3, 31, 40], there has been considerable work on reconstructing hands (for instance with a known shape model like [36]) along with outputs such as poses [12] or 3D meshes [20, 19, 7]. While these approaches can often produce meshes, they usually rely on strong supervision. In contrast, our approach derives its signal from using a small amount of information to make sense of optical flow in videos. While our approach resembles work on human-object relationship detection [45, 17], our relationship is specifically physical contact (e.g., in HOI [45] a person may be using a monitor if they are holding a mouse while we would only segment the monitor if it were physically grasped). Robots accomplish similar object-in-contact tasks [14], but can be challenged by the need to collect large scale interactions. The most related work is understanding objects in contact, ranging from detailed contact models like [5] to bounding boxes like [40]. Our work produces a richer per-pixel segmentation compared to [40] while also requiring less supervision.

We extract this information from optical flow, which has been known as a signal for perceptual organization since the Gestaltists [47] and Gibson [16]. Optical flow at inference time has been used, for instance for doing motion segmentation ([41, 6, 35, 26, 11, 25] among many others). When coupled with learning machinery, the optical flow provides a supervisory signal that can be absent at inference time; our work falls into this category. This signal has been used to learn representations e.g., for pure recognition [34], but our approach is most related to its use for perceptual organization, for instance learning correspondence [46, 24, 48] or boundaries [50, 27]. Our work is most similar to works learning boundaries but focuses on a specific type of boundary – those of hand-held objects. Rather than learning to segment everything, we learn to group hands and the objects they hold. To the best of our knowledge, there is no work that does this; the closest is [4], that finds important objects for an egocentric user based on gaze (as opposed to what object is in a person's hand).

Our approach for learning uses attention for prediction and contrastive learning for grouping. We use attention as a means of relating query points against an entire image. This mechanism is common in methods that solve visual question answering [1, 30], though our use of attention aims to instead discover unknown object extents. In this regard, our work is loosely related to [49], an attention-based visual question answering method. Contrastive learning has become a standard formalism for self-supervised learning, including remarkable performance [8, 33, 2]. We use a formulation inspired by [44], which uses saliency as a grouping signal on Internet image data. In our method, we use optical flow, hands and people instead. Our approach also combines self-supervised learning for per-pixel embeddings with supervision that aims to directly predict which object goes with the hand.

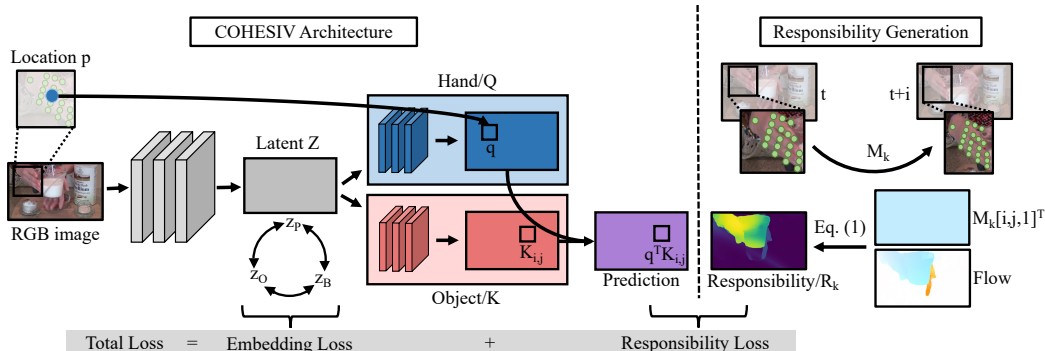

Figure 1: *COHESIV* architecture (left) and responsibility generation (right). First, a homography capturing hand movement is fit between hand vertices in frame t and hand vertices transformed by flow from t to t+i. Then, responsibility for each hand and the background is computed by Equation 1. *COHESIV* takes an RGB image and hand location p as input, uses a UNet to get embeddings for each pixel, conducts contrastive losses between objects, people, and background on the embeddings, and finally inputs embeddings to hand and object branches to relate query points to held-objects in producing a predicted responsibility mask.

## 3 Method

Our approach learns to map the image and an on-hand query point to a prediction of what parts of the scene move with the hand at the query point. While the system requires a single image at test time, it makes use of the rich temporal signal found in video for training. At the core of our approach is the notion of *responsibilities* for hands, inspired by the notion of responsibilities when fitting a GMM [15]: the responsibility of a pixel for a hand is how well that hand explains the pixel's motion compared to other hands and the background (Section 3.1).

At test time, we aim to predict the hand and hand-held object masks for the queried hand. Our network (Section 3.2) predicts this by mapping the image to a shared embedding, which is in turn converted into per-pixel object features that are attended to by per-pixel hand detectors. At training time, the responsibilities provide the basis (along with the pixel locations of humans) for a set of losses (Section 3.3) that we use to train our network, using both direct supervision on the outputs with responsibility maps as well as a contrastive loss on the embeddings.

Throughout, we assume access to systems that can estimate between-frame optical flow, landmarks on the hand, and which pixels belong to people. We use optical flow from RAFT [43], projected hand joints from FrankMocap [37] (to get a set of points on the hands), and people masks from an off-the-shelf people segmentation system [23].

### 3.1 Responsibility

We formalize the notion of synchronous motion (or common fate [47]) for hand and in-hand object via the notion of *responsibility*. Given an optical flow map $\mathbf{O} \in \mathbb{R}^{H \times W \times 2}$ and a set of $N$ hands $\mathcal{H}$, we aim to produce $N$ responsibility maps $\mathbf{R} \in \mathbb{R}^{H \times W \times (N+1)}$ with $\sum_{k=1}^{N+1} \mathbf{R}_{i,j,k}=1$ that explain how well each pixel is explained by each hand's motion model or the background. For the kth hand, we formally model the responsibility as a temperature-softened softmax per-pixel, or:

$$\mathbf{R}_{i,j,k} = \frac{\exp_t\left(-d_k(\mathbf{O}_{i,j,:})\right)}{\exp_t\left(-d_{\mathrm{BG}}(\mathbf{O}_{i,j,:})\right) + \sum_{k'=1}^{N}\exp_t\left(-d_{k'}(\mathbf{O}_{i,j,:})\right)} \tag{1}$$

where $\exp_t(x) = \exp(-x/t)$ is an exponential with a tunable temperature (set to 2 based on a small held-out set) and $d_{\mathrm{BG}}$ and $d_k$ compute distances between an optical flow vector and a model (namely a background model and a model for the kth hand respectively). Equation 1 requires building and evaluating distances between the movement seen in a flow vector $\mathbf{o} \in \mathbb{R}^2$. We treat the background as static and thus the model $d_{\mathrm{BG}}(\mathbf{o})$ is simply $||\mathbf{o}||_2$. This is not the best model for egocentric data, but we find it to be effective, potentially due to the high frame rate of the dataset [9] we use.

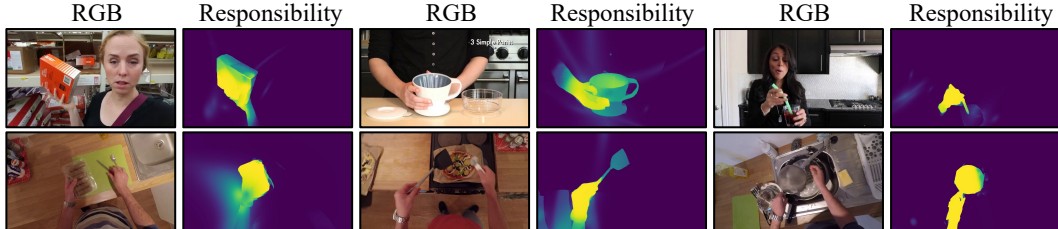

| RGB | Responsibility | RGB | Responsibility | RGB | Responsibility |

Figure 2: Responsibility maps. Images and responsibility maps from 100DOH (top row) and EPICK (bottom row).

Our hand models $d_k(\mathbf{o})$ assume a *planar* motion because it is simple to solve for, can handle out of image-plane rotation, and is a reasonable approximation at the distances and timescales the data shows. This entails fitting a homography $\mathbf{M}_k \in \mathbb{R}^{3\times3}$ on hand landmarks. Supposing proj$(\cdot)$ converts homogeneous to normal coordinates ($[x, y, z] \rightarrow [x/z, y/z]$), then the landmarks should satisfy $[i + \mathbf{o}_1, j + \mathbf{o}_2]^T = \text{proj}(\mathbf{M}_k[i, j, 1]^T)$ for a pixel $i, j$ with flow $\mathbf{o}$. We tried with a variety of options, but found simply fitting $\mathbf{M}_k$ to the estimated flow at FrankMocap [37] vertices to be most effective. Using [37] to provide correspondence proved worse, likely since the resulting hand motion model also included disagreement about the landmark locations in each frame. Then, given a model $\mathbf{M}_k$ and pixel $i, j$ with flow $\mathbf{o}$, the models' prediction is proj$(\mathbf{M}_k[i, j, 1]^T)$. We define the distance as the difference between the actual flow and the predicted flow,

$$d_k(\mathbf{O}_{i,j}) = \left|\left|[i, j]^T + \mathbf{O}_{i,j} - \text{proj}(\mathbf{M}_k[i, j, 1]^T)\right|\right|_2. \tag{2}$$

While simple, the planar model has a counter-intuitive catch. By handling out of plane rotation, the planar model will produce many points that match optical flow by accident specially for static cameras: proj$(\mathbf{M}_k[i, j, 1]^T)$ often has zeros in the image. While a learning system could learn to ignore these accidental matches by training on a large dataset, we accelerate this by averaging it out: we calculate responsibility maps to the next and previous 3 frames, and then average them.

**Techniques:** We estimate optical flow with RAFT [43] and hand vertices with FrankMocap [37]. FrankMocap fuses a hand detector [40] with a network that predicts 2D and 3D hand joints. This information is used to compute homographies and then responsibility maps. Once responsibilities are computed, they serve as pseudo-labels at training time and are then discarded. At test time, we aim to predict hands and held-objects for new images.

**Data Selection and Implementation Details:** The data we use is challenging non-scripted videos that include vast amounts of non-interaction. We select our training and validation data so that the network primarily sees hands that are visible and in contact. We select clips where hands appear in 10 consecutive frames as well as hands are moving within the clip. In 100DOH, the clips are detected as visible hands holding reasonably sized objects using [40]. We note that we only have to do this selection since the data used is large and non-scripted. In EPICK, we use their action segments. (see the supplemental for more details)

## 3.2 Architecture

Our approach aims to take an image $\in \mathbb{R}^{H\times W\times 3}$, a query point $\mathbf{p} = [x, y]$, and produce a segmentation of the hand and object that the hand at $\mathbf{p}$ is in contact with. Our architecture produces a query feature map $\mathbf{Q} \in \mathbb{R}^{H\times W\times A}$ representing per-hand detectors and a key feature map $\mathbf{K} \in \mathbb{R}^{H\times W\times A}$ representing object features. At test time, one can extract $\mathbf{Q}_{x,y,:}$ and produce a detection score at any $i, j$ by $\mathbf{q}^T \mathbf{K}_{k,l,:}$. Thus, a forward pass can be used to parse multiple hands by extracting different locations.

We produce $\mathbf{Q}$ and $\mathbf{K}$ from a common backbone that produces an embedding $\mathbf{Z} = \mathbb{R}^{H\times W\times F}$. We predict the embedding $\mathbf{Z}$ with a standard U-Net-style [38] network with a SE-Net [22] (se-resnext50-4d) backbone and ImageNet [39] pretrained encoder weights. Then, we use two lightweight paths (two $3 \times 3$ Conv layers) from $\mathbf{Z}$ to heads $\mathbf{Q}$ and $\mathbf{K}$ (see the supplemental for a full description).

The use of an intermediate embedding plus the attention heads enables the heads to handle integration of hand-specific information (e.g., given this hand, this particular object goes with it) while the embedding captures how people, object, and background pixels are distinct. The asymmetry via two

heads also enables producing the *object* the hand is holding rather than only the hand: for a single embedding, the hand must have maximum self-similarity if the data is normalized.

**Hand Detector:** One can obtain $\mathbf{p}$ via a hand detector, but this detector can also be built into *COHESIV*. We have trained a hand detector on a frozen backbone up to $\mathbf{Z}$, enabling it to run entirely on its own. We trained another head (similar to head $\mathbf{Q}$ and $\mathbf{K}$, see the supplemental for details) to instead map the latent space to a 1D heatmap for hand center location. Like pose estimation techniques, this model is trained to minimize the MSE to a Gaussian placed at the center of the hand box reported by [40].

**Inference:** At inference time, the model processes a single image to return embeddings that separate people, objects, and background. When optionally given a query point on the hand, it also predicts the hand and object segmentation masks for that query, enabling an agent to investigate any potential point of interest. For objects, we first average 500 embeddings produced from training images wherever our pseudo-labels are "object". We then calculate a cosine similarity between this "average object embedding" and the predicted embedding for the inference image. We take this similarity map and the predicted responsibility attention map (i.e., $\mathbf{q}^T\mathbf{K}_{i,j}$) and fit two thresholds on validation data, such that object pixels need to both have "objectness" above a threshold and predicted responsibility above another threshold. Hand mask predictions are created analogously, except an "average person embedding" is created from "person" pseudo-labels and used in combination with an embedding and responsibility threshold.

## 3.3  Losses

We train with two losses that encourage different behaviors for the embedding and attention heads: the responsibility loss direct optimizes the output while the embedding loss encourages pixels on hand-held objects to group together. Both require dot products between a full embedding space and a handful of vectors as opposed to an expensive all-pairs $H^2W^2$ tensor. While progress on both losses is often in the same direction, optimally minimizing the proposed losses on the same feature is impossible with multiple hands – the embedding loss is satisfied if all hand-held objects are identical while the responsibility loss is not satisfied until each hand-held object is separated. We therefore jointly minimize these losses on *different* features.

**Responsibility Loss:** Our first loss directly supervises responsibility: given a point $\mathbf{p} = [x, y]$ on a hand (blue dot in Figure 3), we compute $\mathbf{Q}^T_{x,y,:}\mathbf{K}_{i,j,:}$ for all $i, j$ and train the network to match the responsibility. After computing the prediction, we clamp it to [0,1], meaning that the system can make an object not match a query point by making the vectors orthogonal (a large and vast space) rather than polar opposites (a specific location). We minimize a per-pixel Smooth-L1 loss (with $\beta = 0.5$) between the predicted value and the computed responsibility map.

**Embedding Loss:** We also use a contrastive loss, based on [44], that encourages coherent embeddings regardless of hands. These use a set of masks for people, objects, and background via mask operations on [23] and the union of responsibilities. Suppose $\mathbf{z}_P$, $\mathbf{z}_O$, and $\mathbf{z}_B$ represent the average embeddings from $\mathbf{Z}$ for the people, objects, and background masks. Then the contrastive loss aims to ensure that image embeddings within these masks are all similar to one another, and that the average embedding across each of these masks is far from the average embedding of the other two groupings. The contrastive loss for hand-held objects at pixel $i, j$ is the negative log-likelihood

$$-\log\left(\frac{\exp\left(\mathbf{z}_O^T\mathbf{Z}_{i,j}\right) + \epsilon}{\exp\left(\mathbf{z}_O^T\mathbf{Z}_{i,j}\right) + \exp\left(\mathbf{z}_P^T\mathbf{Z}_{i,j}\right) + \exp\left(\mathbf{z}_B^T\mathbf{Z}_{i,j}\right) + \epsilon}\right),\tag{3}$$

where $\epsilon = 10^{-3}$ aids numerical stability. This loss is evaluated only for pixels $i, j$ that are members of the hand-held object pseudo-label. The same loss is analogously defined for people and background, respectively replacing $\mathbf{z}_O^T$ in the numerator for $\mathbf{z}_P^T$ or $\mathbf{z}_B^T$ and evaluated only for pixels $i, j$ in the person or background pseudo-label.

**Implementation:** We weight the attention loss and embedding loss as 10:1. People masks are outputs from [23]; object masks are responsibility masks that have had the person segmentation subtracted from them, and thresholded; background masks are the remaining pixels.

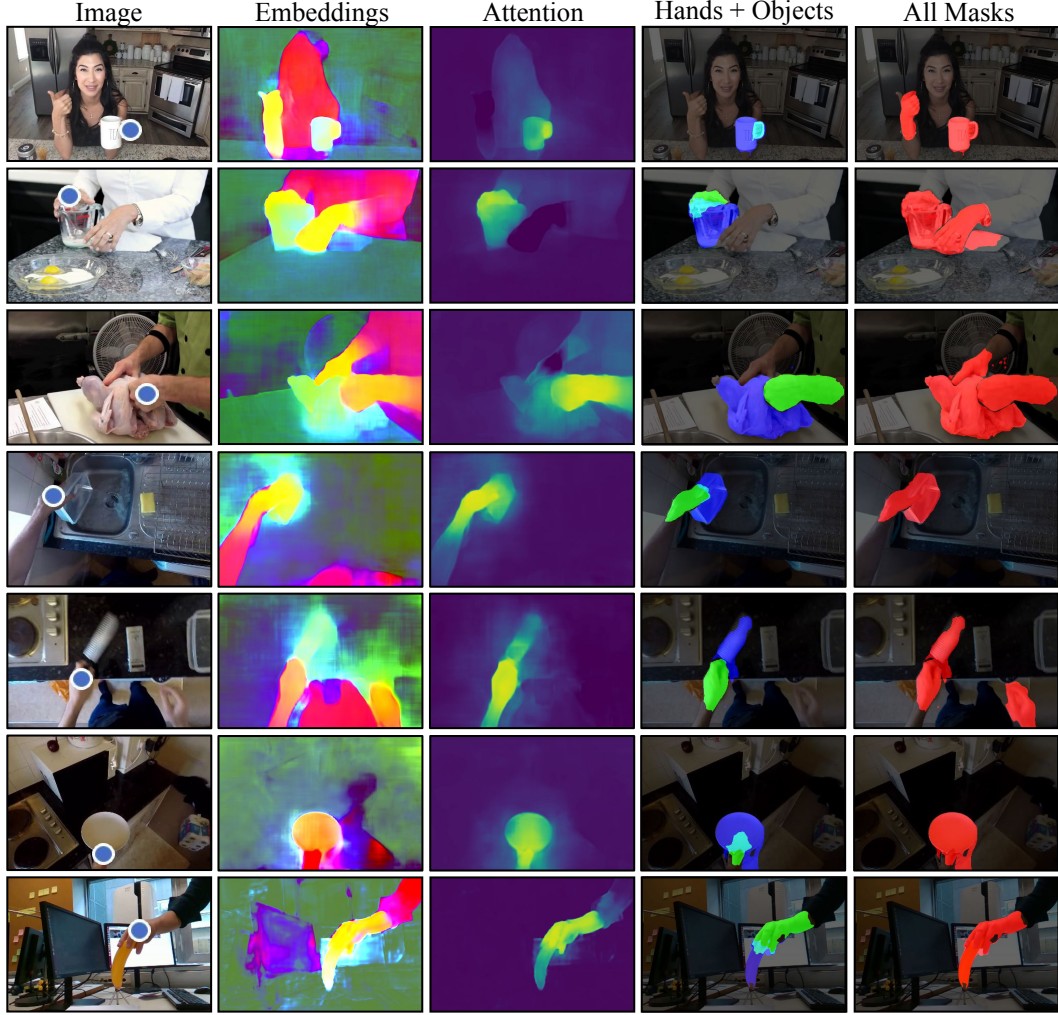

| Image | Embeddings | Attention | Hands + Objects | All Masks |

Figure 3: Visualizing *COHESIV* on 100DOH (rows 1-3), EPICK (rows 4-6), and HO3D (row 7). Given the RGB image and on-hand query point (blue dot) as input (col 1), the embeddings column (col 2) shows the contrastive visual embeddings visualized using PCA dimensionality reduction ($d = 3$); the attention column (col 3, viridis colormap) shows the predicted responsibility of each pixel's movement related to the query point; the hands and objects column (col 4) shows the predicted segmentation masks for the given pair (green for hands, blue for objects, and cyan for hand object overlap); and the all masks column (col 5) shows all predicted segmentation masks in red.

## 4 Experiments

We evaluate our approach on contact segmentation. Given an image and query point $(x, y)$ on a hand, our goal is to find the extent of the hand and potential in-contact object. We test on 100DOH [40], EPICK [9], and HO3D [18]. We first compare against baselines and alternative methods and then ablate model components to understand how design decisions impact performance. We qualitatively evaluate our methods and show how learning per-pixel embedding helps segmentation. Finally, we evaluate a hand detector branch that makes the model usable without a pre-detected hand location.

### 4.1 Experimental Setup

**Datasets:** We conduct experiments on 100DOH [40], EPICK [9], and HO3D [18]. We first generate 10-frame clips from these video datasets following the original dataset splits. Given each 10-frame clip, we compute and select responsibility maps from the middle ($4th$, $5th$, $6th$ and $7th$) frames

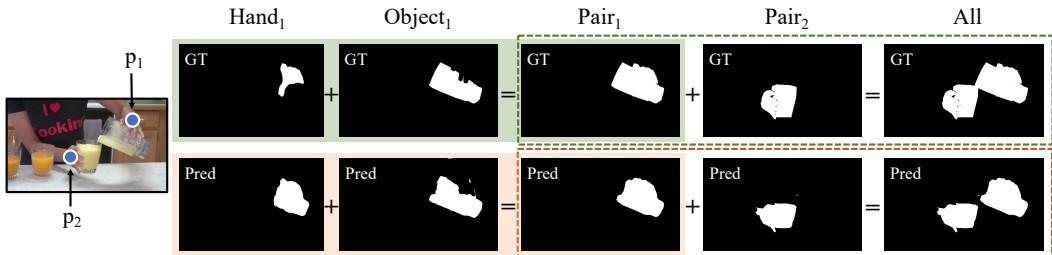

Figure 4: Four evaluation tasks. The first row illustrates that a **hand** and **object** annotation are added to form a **pair** annotation. Then all pairs in an image are added to form an **all** annotation. *COHESIV* inputs the RGB image on the left and query point $p_1$, producing outputs shown in the bottom row for $Hand_1$ and $Object_1$, which are then combined into $Pair_1$ before finally all pairs are combined into All.

for use as training data (see supplemental for details). For each hand, we choose a query point by averaging the vertices produced by the 3D hand mesh prediction in [37].

In the 100DOH [40] dataset, we generate 80.4K 10-frame training clips from the "make drinks" and "make food" genres. 100DOH [40] is collected from a large variety of public YouTube videos containing hand and object interaction from everyday life with offensive content intentionally removed. Specifically, we select from the "make drinks" and "make food" videos due to the variety of objects and the tendency of these objects to be roughly rigid. In EPICK [9], we generate 30K 10-frame training clips from action segments. EPICK [9] is an egocentric video dataset that captures daily activities in the kitchen and only shows peoples' hands and home environments (also without offensive content). The HO3D [18] dataset contains sequences of people manipulating different objects in laboratory settings, and has ground truth hand and object segmentation masks for evaluation. We generated 8K clips from HO3D, using 5K of these clips for training.

**Annotations for Evaluation:** To evaluate our approach on the 100DOH and EPICK datasets, we annotate hand and held object masks with a 2-step manual labeling task via a labeling platform. Given a red dot on each hand, we ask annotators to label the hand mask; then, given the hand mask, we ask annotators to label the in-contact object mask if it exists. This gives 1,123 test and 482 validation images for 100DOH [40] and 1,169 test and 437 validation images for [9].

**Training Details:** We trained the *COHESIV* model and all ablations on 5 GTX 1080 GPUs. We use the AdamW [29] optimizer, with initial learning rate of $10^{-2}$ and batch size of 10. We used a learning rate scheduler that cut the learning rate in half after 5 epochs without a validation loss decrease. We used early stopping and halted training if the validation loss did not decrease after 10 epochs.

**Evaluation Metrics:** We evaluate using Intersection over Union (IoU) on four different tasks that change the definition of ground-truth positive: **(1) All:** any hand or held object is positive, which evaluates how well a method can identify all hands and hand-held objects (akin to semantic segmentation); **(2) Pair:** the hand in question and any object it holds are positive, which evaluates how well a system can individuate hands and their held object (akin to instance segmentation); **(3) Hand:** only the hand in question is considered positive; **(4) Object:** only the object the hand is holding (if any) is considered positive. *COHESIV* can solve all four tasks. Other methods are sometimes unable to produce all 4 tasks, which we denote with "-". We calibrate thresholds for binarization using grid search on the validation set.

## 4.2 Contact Segmentation

We use this setup to compare our approach qualitatively and then against baselines. We then ablate the model and test the automatic detection system built on *COHESIV*.

**Qualitative Results:** Figure 3 visualizes *COHESIV* outputs. Our system has the ability to distinguish hands via the query point. The queried hand is of high probability while the unrelated hand is of low probability. At times, large objects are poorly predicted, with predicted responsibility decreasing for pixels at points distant from the hand. In such instances, using the embedding to separate objects from people and the background is helpful.

Table 1: Comparing *COHESIV* against 3 baselines and a supervised method on the 100DOH and EPICK datasets using mIoU (%). Our proposed model outperforms the 3 baselines and is comparable to the supervised bounding box method.

| | 100DOH [40] | | | | EPICK [9] | | | |
|---|---|---|---|---|---|---|---|---|
| | All | Pair | Hand | Obj | All | Pair | Hand | Obj |
| COHESIV | 51.9 | 46.5 | 53.6 | 29.3 | 43.2 | 42.1 | 60.7 | 19.5 |
| Saliency [51] | 25.2 | 20.1 | 8.6 | 17.0 | 21.6 | 15.9 | 6.0 | 11.7 |
| Flow [43] | 29.3 | 21.5 | 12.9 | 12.1 | 15.4 | 11.9 | 6.2 | 6.6 |
| Thresholded Responsibility | 44.5 | 37.0 | - | - | 42.9 | 30.0 | - | - |
| Supervised Bounding Box [40] | 56.9 | 47.0 | 56.5 | 34.9 | 54.3 | 44.8 | 53.8 | 34.4 |

Table 2: Ablations of *COHESIV*, evaluated on the 100DOH and EPICK datasets using mIoU(%). It is worth noting that the solely attention based model cannot readily separate the co-occurring motion of hand and objects.

| | 100DOH [40] | | | | EPICK [9] | | | |
|---|---|---|---|---|---|---|---|---|
| | All | Pair | Hand | Obj | All | Pair | Hand | Obj |
| COHESIV | 51.9 | 46.5 | 53.6 | 29.3 | 43.2 | 42.1 | 60.7 | 19.5 |
| Attention-Only | 42.8 | 40.0 | - | - | 38.1 | 37.8 | - | - |
| Embeddings-Only | 25.7 | 18.3 | 13.2 | 22.9 | 30.0 | 20.8 | 24.6 | 14.4 |
| COHESIV w/ ResNet Backbone | 45.8 | 41.2 | 48.1 | 25.2 | 39.8 | 39.1 | 55.2 | 17.9 |
| COHESIV w/ Predicted Query | 47.7 | 42.8 | 47.8 | 28.1 | 40.0 | 38.6 | 55.1 | 19.4 |

Table 3: Comparing *COHESIV* against 3 baselines and a supervised method on the 100DOH and EPICK datasets using mIoU (%). Our proposed model outperforms the 3 baselines and is comparable to the supervised bounding box method.

| | 100DOH | | | | EPICK | | | |
|---|---|---|---|---|---|---|---|---|
| | All | Pair | Hand | Obj | All | Pair | Hand | Obj |
| COHESIV | 51.9 | 46.5 | 53.6 | 29.3 | 43.2 | 42.1 | 60.7 | 19.5 |
| Saliency | 25.2 | 20.1 | 8.6 | 17.0 | 21.6 | 15.9 | 6.0 | 11.7 |
| Flow | 29.3 | 21.5 | 12.9 | 12.1 | 15.4 | 11.9 | 6.2 | 6.6 |
| Thresholded Responsibility | 44.5 | 37.0 | - | - | 42.9 | 30.0 | - | - |
| Supervised Bounding Box | 56.9 | 47.0 | 56.5 | 34.9 | 54.3 | 44.8 | 53.8 | 34.4 |

**Comparisons:** To understand the capabilities of *COHESIV*, we compare it against 3 baselines and a supervised method.

**(1) Saliency**: We use a recent saliency detection system [51] which aims to identify regions of interest in an image, and it serves as a benchmark for how foreground/background segmentation approaches might perform on our task. **(2) Flow**: We use RAFT [43] to estimate the flow between adjacent pairs of frames in the video, then fit a threshold across validation annotations and treat all flow above this threshold as a positive prediction. **(3) Thresholded-Responsibility**: We directly binarize the responsibility pseudo-labels using a threshold chosen from the validation set. **(4) Supervised Bounding Box**: Rectangular areas from the supervised hand-object bounding box detector in [40]. As it is trained on over 132K annotated samples from the training sets of both 100DOH [40] and a variety of egocentric datasets, including [9], we do not expect to match its performance. We report it as a comparison, however, for context. Of course, by virtue of our system producing segmentations, we can in principle get refined segmentations (although a box is often a good approximation for the object).

The **Saliency** and **Flow** baselines aim to serve as alternate explanations for the results seen (e.g., that the approach is just picking up on a salient object or moving pixels), both of which are generated for the entire image and then evaluated directly against all four targets (All/Pair/Hand/Object).

Table 4: Ablations of *COHESIV*, evaluated on the 100DOH and EPICK datasets using mIoU(%). It is worth noting that the solely attention based model cannot readily separate the co-occurring motion of hand and objects.

| | 100DOH | | | | EPICK | | | |
|---|---|---|---|---|---|---|---|---|
| | All | Pair | Hand | Obj | All | Pair | Hand | Obj |
| COHESIV | 51.9 | 46.5 | 53.6 | 29.3 | 43.2 | 42.1 | 60.7 | 19.5 |
| Attention-Only | 42.8 | 40.0 | - | - | 38.1 | 37.8 | - | - |
| Embeddings-Only | 25.7 | 18.3 | 13.2 | 22.9 | 30.0 | 20.8 | 24.6 | 14.4 |
| COHESIV w/ ResNet Backbone | 45.8 | 41.2 | 48.1 | 25.2 | 39.8 | 39.1 | 55.2 | 17.9 |
| COHESIV w/ Predicted Query | 47.7 | 42.8 | 47.8 | 28.1 | 40.0 | 38.6 | 55.1 | 19.4 |

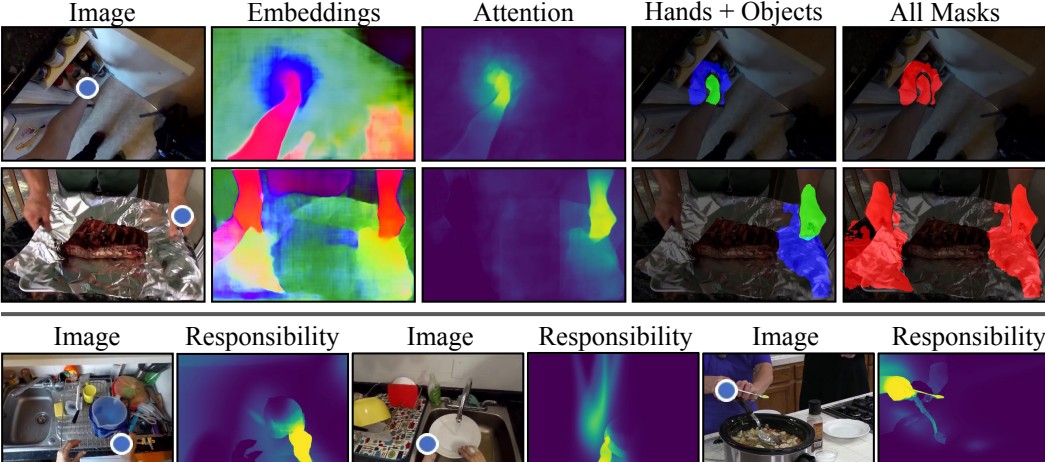

Figure 5: Two types of failures. The first row shows the prediction of an amorphous halo around the hand rather than the grasped bottle. The second row shows a missing interior to the grasped tinfoil tray. The last row shows three failures of our computed responsibility maps related to distance to the hand, hand occlusion, and accidentally similar motion between adjacent objects.

**Ablations and Extensions:** We include a number of modifications to *COHESIV* to understand the importance of various components in Table 4. We test models that train with only attention loss or embedding loss, followed by one that uses a ResNet backbone rather than SE-NET [22]. Finally, we test an extension that uses a predicted hand point from our system rather than locations from an existing hand detector [37] as mentioned in Section 3.2.

**Quantitative Results:** In Table 3, we present how our model compares to similar baseline methods. Our method approaches the performance of the supervised approach in [40] and surpasses it on the EPICK Hand task. Our systems outperform simpler approaches which find salient objects or use optical flow estimations above a threshold. We conducted a limited evaluation on the HO3D dataset (where All is the same as Pair due to the absence of multiple hands). Our approach obtains good performance: 49.1 (Pair), 49.6 (Hand), 30.6 (Obj), just 8.4 (Pair), 4.2 (Hand), and 8.5 (Obj) points short of Supervised Bounding Box [40].

In Table 4, we demonstrate the value of our hybrid attention and embedding model, by comparing against ablated versions. Jointly training with both generally improves results on 100DOH and EPICK. Our extended hand-detection branch works slightly worse than using an existing hand detector [37], but shows comparable performance and enables a single forward pass.

## 4.3 Analysis and Limitations

While we cannot directly evaluate the learned embeddings, we show results throughout the paper using PCA. We fit PCA on the embeddings, $H \times W \times F$ (dimension $F = 64$ here) per-pixel feature vectors, and convert the resulting dimensions into RGB space after z-scoring.

Our system is by no means perfect and we show some failure modes in Figure 5. One is in our system. As shown in the first row, our model find a larger and amorphous blob but instead of the detailed grasped bottle. The second row shows that for large non-rigid objects, it is hard to capture the whole area. The other one is in the responsibility. Generally, when the motion is very small and affected by shadow and occlusion, it is hard to get a good responsibility map. The third row shows the responsibility map quality is affected when the object part is distant from the hand (less similar motion with the hand), when the hand is heavily occluded, and when there are other objects in the image move similarly with the hand.

In addition to standard mispredictions by deep learning systems, some failures are intrinsic to the signal we use: if the object is unlikely to move, then our system is unlikely to see it as a hand-held object. We see this as a challenging next step, and a problem of great importance.

## 5  Discussion

We present a system for segmenting hands and hand-held objects by learning from motion in videos through weak supervision. We generate responsibility maps as pseudo-labels for learning and incorporate contrastive learning to distinguish held objects from people and background in images. Our work and possible extensions, such as work that separates the embeddings of objects classes, make possible research that can readily address the long-tail of object classes that exist in the real world, outside of prescribed datasets. Our method an attempt to learn about objects from hands using common motion signals. We hope it serves as a useful next step in developing a deeper understanding of the interrelationship between hands and objects.

**Broader Impact:** Systems that learn from signals or correspondence within videos (rather than manual labels from human) have the potential to see and discover portions of the visual world that have been overlooked by researchers. Using hands as a medium to explore adjacent objects in the natural world is a promising way to use manipulation to learn physical relations. Starting from hands and generalizing to the unlabelled object segments of the world can positively impact many downstream applications that need a more real world understanding of object classes than is provided by a fixed number of classes.

On the other hand, our system can have negative downstream consequences. First, while the data in public videos is often closer to reality than Flickr images, deep learning systems can replicate and amplify the biases observed in these videos. While there has been lots of effort towards making datasets more reflective of everybody, there is still a lot of progress to be made. This requires both careful reflection during data curation and monitoring during deployment. Second, premature deployment of these systems without proper oversight can have negative affects, especially for people not well-represented.

**Acknowledgments:** We thank Mohamed El Banani, Karan Desai, Sarah Jabbour, Nilesh Kulkarni, Max Smith, and Shengyi Qian for the feedback and support. This material is based upon work supported by the National Science Foundation under Grant No. 2006619.

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
