# A    COHESIV: Supplemental Appendices

In Section C, we explain how we filter video clips to use in training, how we compute and filter responsibility maps, and how we annotate validation and test data for both 100DOH and EPICK. In Section D, we describe the network backbones we use, as well as the differences between experimental settings and training. In Section E, we introduce more results including both selected and random results from test data.

# B    Checklist Questions

**3a Did you include the code, data, and instructions needed to reproduce the main experimental results (either in the supplemental material or as a URL)?** We misunderstood section 3(a) of the submission checklist. This section asks for a promise to "include the code, data, and instructions needed to reproduce the main experimental results". We mis-read this section as code, data, *OR* enough implementation details to reproduce our experiments. Unfortunately, the dataset used is proprietary, and not ours to distribute. Though we cannot release the dataset, we here-in provide the necessary details to replicate our experiments and will release the annotations and code publicly if accepted.

**4e Did you discuss whether the data you are using/curating contains personally identifiable information or offensive content?** In EPICK, no images of faces or individual are included. The captured footage will only include recordings of the hands during object interactions with no offensive content. 100DOH is a large Internet-scale dataset consisting of already publicly available footage. The creators of 100DOH claim to have already removed offensive content. We checked the data we sent to thehive and found no offensive content.

**5c Did you include the estimated hourly wage paid to participants and the total amount spent on participant compensation?** Estimating the hourly wage is difficult since we worked with an intermediary crowdsourcing company that uses adaptive techniques We paid approximately $0.15 per segment, totally $3,106.61.

# C    Data Processing

## C.1    Video Clips

We next describe the process by which we select video clips for use in the training and validation of our models.

**100DOH Clips:** For the 100DOH dataset, we use videos from the "make food" and "make drink" classes to generate clips. Due to effects from camera pose and a diversity of hand sizes and configurations (compared against egocentric hand videos), we set constraints on the tracked hand in each clip. Specifically, we choose clips that have a: (1) stationary background from the shot detection; (2) one tracked hand appearing in 10 consecutive frames; (3) reasonably sized and centered tracked hands (between $5\%$ and $50\%$ of the image diagonal width and not within $5\%$ of the image margin); (4) in contact and moving (tracked hands in contact with objects for $> 2$ frames and move $> 2\%$ of the image diagonal). Each clip can have at most a $50\%$ overlap in frames. We inherit the original train/val/test split from 100DOH.

**EPICK Clips:** We generate clips from action segments in the EPICK dataset. In each action segment, we randomly pick a start frame and choose one out of every three frames until we get to 10 kept frames. We inherit the original train/test split. We only use the unseen participants testset from EPICK. By using only the unseen participants testset, we ensure the model only tests on people it has never seen before. Since there is no validation set in the dataset, we use the first video of each participant (i.e., $PXX\_YY$ where $YY = 01$) from the trainset as validation videos for generating validation clips.

## C.2    Video Filtering

Prior to training on EPICK and 100 DOH, we filter the training samples used for the Embedding and Embedding + Attention models to exclude images where any of the following properties hold:

- Either $< 5$ people pixels total, or entirely people pixels, as predicted by Ternaus.
- Either $< 5$ responsibility above a threshold pixels total, or entirely responsibility above a threshold pixels, as computed by our method.
- Images where people or responsibility estimates contain NaNs.
- Images where there are no background pixels, as the entire image is either people or responsibility.

These conditions filter out rare edge cases of degenerate masks that make loss calculations impossible and which break training.

### C.3 Selecting Responsibility Maps

Next, we describe how we choose a subset of our original training set to meets empirical requirements for accurate responsibility computations.

From each 10 frame clip, we first select the 4 frames that have +/- 3 frames previous and subsequent to them. We then compute the necessary features between these +/- 3 frames and the center frame to produce a single responsibility map. Once we have these 6 maps, we fuse them by summing them together and dividing by the maximum value, producing one refined responsibility map. From this fused responsibility map, we clean maps and filter maps from training that do not meet the following requirements:

1. High responsibility around the hand and low responsibility far from the hand: *dist2hand* (sum of responsibility $*$ distance to hand center) in the interval $t1$.

2. Reasonably sized hands: the ratio of the length of the hand bounding box diagonal to the length of the image diagonal is in the interval $t2$

3. Reasonably moving hands: Hand movement is $> t3$ the length of the image diagonal.

4. Responsibility covering objects: the object-hand area ratio is $> t4$, and the *dist2hand* within the $\times 2$ scaled hand bounding box excluding hand area is $> t5$.

5. Image size is $t6$.

**100DOH Responsibility Maps:** We set $t1 = [5, 100]$, $t2 = [10\%, 70\%]$, $t3 = 1\%$, $t4 = 50\%$, $t5 = 25\%$, $t6 = 368 \times 640$.

**EPICK Responsibility Maps:** Since the frame resolution is different from 100DOH, we use different thresholds for the constraints as $t1 = [1.25, 25]$, $t2 = [10\%, 70\%]$, $t3 = 1\%$, $t5 = 25\%$, $t6 = 256 \times 464$.

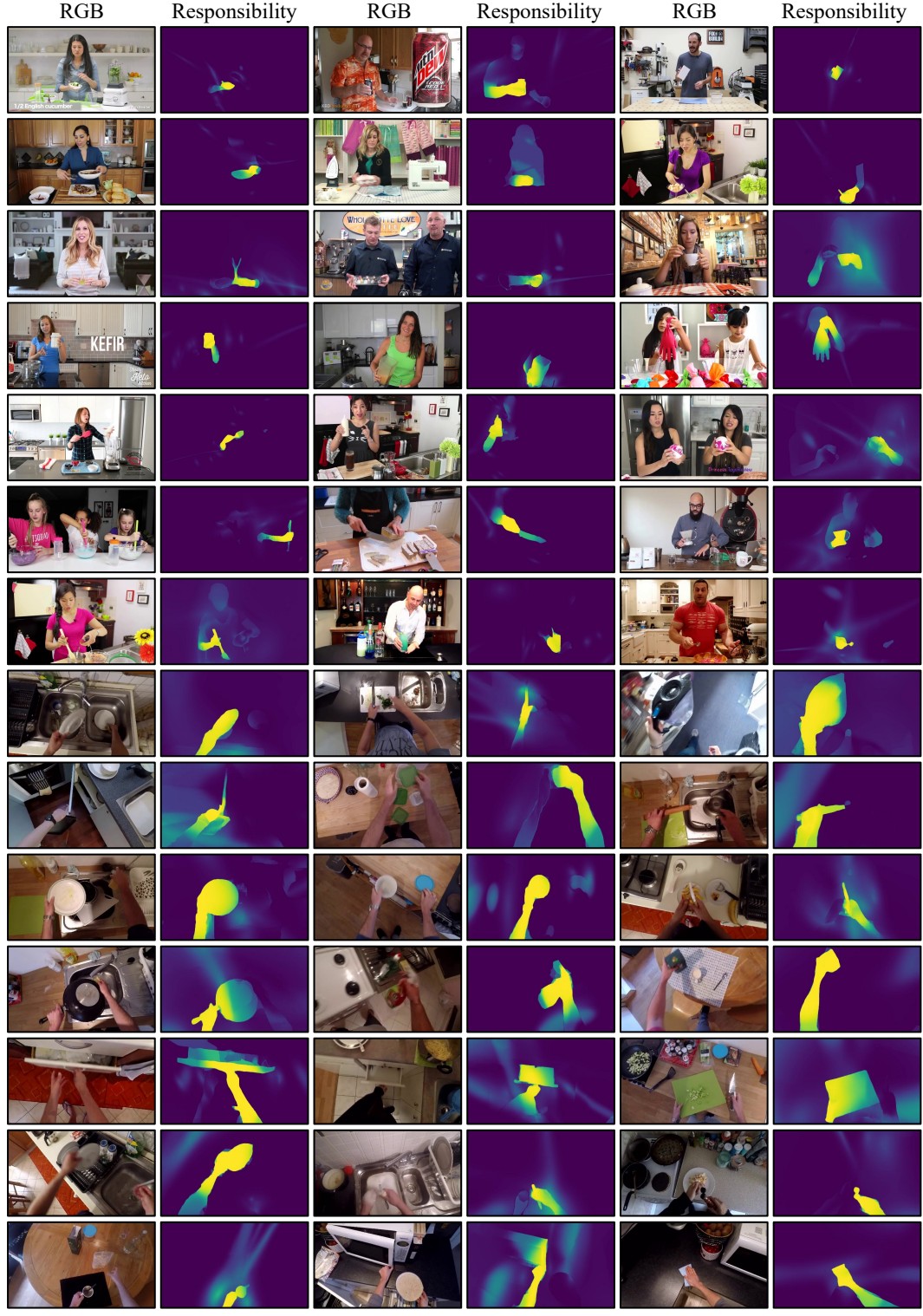

Figure 1: More images and associated single hand responsibility maps on 100DOH (top 7 rows) and EPICK (bottom 7 rows).

Table 1: COHESIV Model architecture. We use a UNet with se-resnext-4d as the encoder backbone to get the embedding of the input. The embedding after normalization is then passed separately through both the query (Q) head and key (K) head, where we choose feature at (x,y) location from Q and do dot product with K. Final results are generated after clipping to the range $[0, 1]$. We use smooth L1 loss (beta=0.5) to supervise training.

| No. | Inputs | Operation | Output Shape |
|-----|--------|-----------|--------------|
| (1) | RGB, (x, y) | Input | $H \times W \times 3, (2)$ |
| (2) | RGB | Backbone: U-Net with SE-Net(se-resnext50-4d) backbone | $H \times W \times 64$ |
| (3) | (2) | Q Head: Conv($64 \rightarrow 64, 3{\times}3$), ReLU, Conv($64 \rightarrow 32, 3{\times}3$) | $H \times W \times 32$ |
| (4) | (2) | K Head: Conv($64 \rightarrow 64, 3{\times}3$), ReLU, Conv($64 \rightarrow 32, 3{\times}3$) | $H \times W \times 32$ |
| (5) | (3)@(x,y), (4) | Attention | $H \times W \times 1$ |
| (6) | (5) | Clipping to $[0, 1]$ | $H \times W \times 1$ |

# D   Architecture

We use a SENet initialized UNet as the backbone for our experiments. We list all details of all network architectures in Table D.

**COHESIV:** Our proposed model architecture is described in Table D. We use a SE-NET backbone (se-resnext50-4d) from (1) with Image-Net pre-trained weights as a backbone to produce image features. This embedding output is then normalized and passed to the query head and key head (each has 2 layers of Conv, using ReLU only with the 1st Conv). Finally, we do attention between the normalized query vector and key features, pass it to a Conv layer, and clip it to the range $[0, 1]$ to get a final predicted responsibility output.

It is worth noting that there are a series of normalizations performed along the embedding dimension. We normalize the embeddings to have a magnitude of 1 after the backbone and after each head and the attention module.

**Attention-only:** This model is also shown in Table D, and the difference to COHESIV is just in the losses applied to the embedding/features that are the result of the backbone network.

**Embedding-only:** The embedding only network is only the first two lines (1) and (2) of the architecture described in D.

## D.1   Inference Details

To transform embeddings and the outputs of attention models into hand and object segmentations, we follow four similar processes for each of the four evaluation targets.

**Objects:** Object predictions are made by first averaging embeddings produced from training images wherever our pseudolabels are "object". We then calculate a cosine similarity between this "average object embedding" and the predicted embedding for the inference image. We take this similarity map and the predicted responsibility attention map (i.e., $\mathbf{q}^T \mathbf{K}_{i,j}$) and fit two thresholds on validation data, such that object pixels need to both have "objectness" above a threshold and predicted responsibility above another threshold.

**Hands:** Hand predictions are made by first averaging embeddings produced from training images wherever our pseudolabels are "person". We then calculate a cosine similarity between this "average person embedding" and the predicted embedding for the inference image. We take this similarity map and the predicted responsibility attention map (i.e., $\mathbf{q}^T \mathbf{K}_{i,j}$) and fit two thresholds on validation data, such that hands pixels need to both have "personness" above a threshold and predicted responsibility above another threshold.

**Pairs:** Pair predictions are made by supplying the network with a query point at inference, predicting the hand and object associated with that query point, and unioning these predicted hand and object masks.

**All:** Inference for all hands and objects is conducted by first predicting each hand/held-object pair. Then all paired-predictions in an image are unioned and this becomes the All prediction.

# E  Extended Results

## E.1  More Results

We present additional qualitative examples to illustrate *COHESIV* in action. For all qualitative results, the first column is the image (with highlighted query point), the second column is the responsibility attention prediction, the third column is the PCA, the fourth column is the clusters, and the final column is the mask.

**Qualitative results on 100DOH** The first row shows a woman holding a container and preparing to scoop items into it. The hand and held object masks are well outlined. Initially, it appears the object mask is erroneously predicted as being atop the hand, however, on second glance the woman is wearing a glove and therefore the overlapping prediction of hand and held object is actually correct. The second rows shows someone playing with playdough on a table. Despite the small size of the playdough, and it's adjacency to the table, the hand and contacted objects are clearly demarcated. The third and fourth rows show an excellent example of someone rolling dough with a rolling pin. Both hands are clearly identified, and the union of the two object masks that split the rolling pin almost entirely identify it. The fifth row shows someone scooping ice cream from a container. The embedding shows a very clear outline of the ice cream. At first glance, the hand and held object mask appears to be poorly predicted, with little coverage of the ice cream. However, upon looking at the all masks, it's clear that the rest of the held object was captured by the predicted responsibility for that hand. The sixth row shows someone serving themselves food, and the embedding identifies potential objects, and although small, the held plate it successfully identified. The seventh row shows someone holding a box on top of a guitar-shaped cutting board. Both hands partly split the object responsibility for the box. The eighth row shows someone holding a roll of paper towel, though only half of it is captured by the detected object mask. The ninth row shows a women holding a pot of food. Both hands successfully split the held object, despite them being small and the pot being large. The final row shows a woman opening a coconut. The embedding is very crisp for the coconut, and the hands both split the resulting object responsibility.

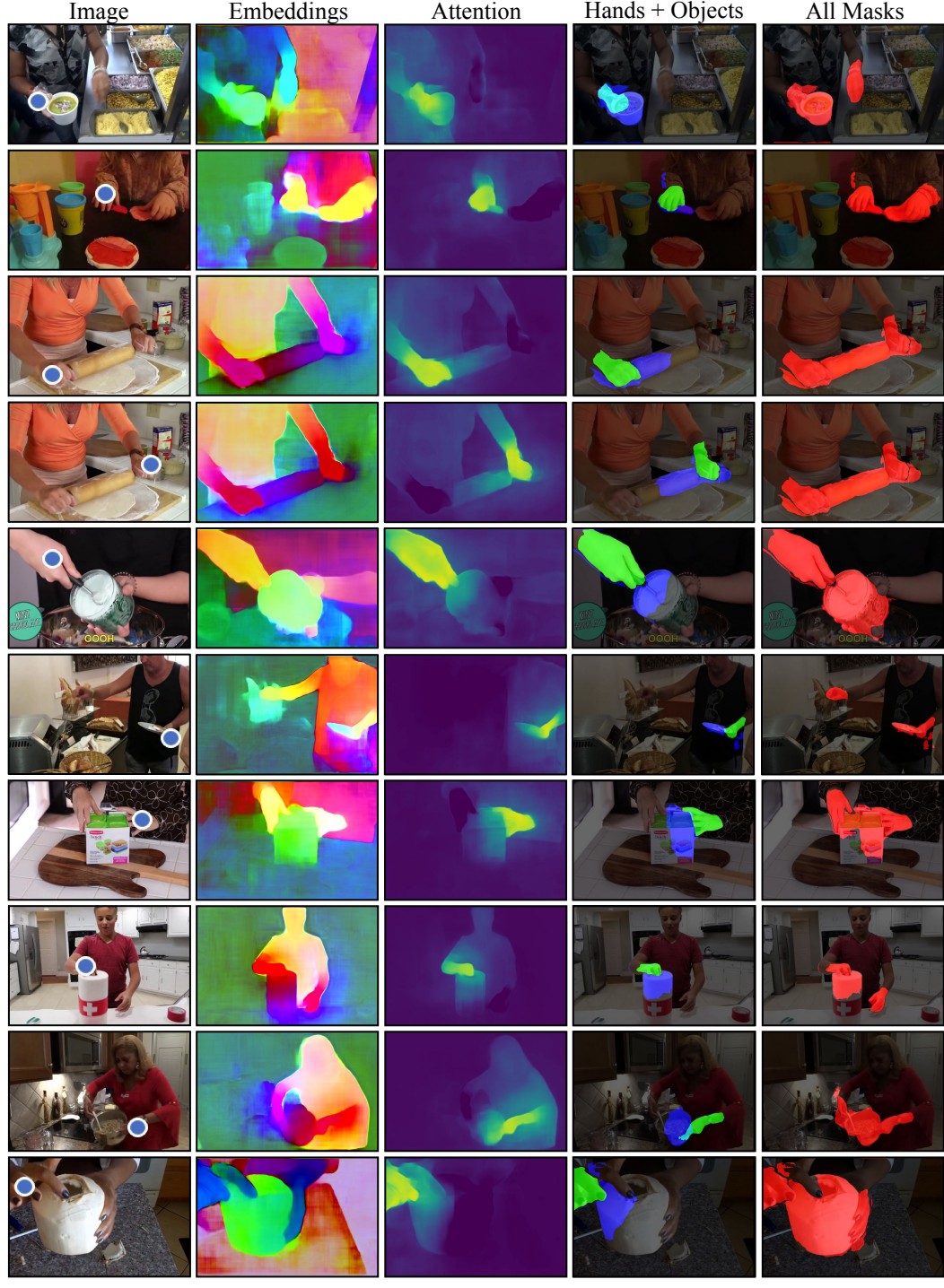

Figure 2: Additional selected examples of model output chosen from 100DOH. The first column shows the image and input query point, the second column is a visualization of the model's embedding, the third column is the predicted responsibility, and the fourth and fifth columns show predicted hand/obj and all masks.

**Qualitative results on EPICK** In the first row a person is stirring vegetables in a pan. The embedding highlights that both the wooden spoon and the vegetables are of interest, while the predicted responsibility shows an interest in the spoon as well. The hand and object predicted masks successfully distinguish that the cook is holding the spatula. The second row shows someone touching an oven mitt. The oven mitt embedding is clearly demarcated compared to the background. The responsibility map for the left hand shows good identification of the object of interest but coverage is not complete. The predicted hand and object mask shows that the right hand is not clearly in contact with the mitt and the predicted responsibility mask is similarly restricted. The third row shows someone washing a colander in the sink. The embeddings are not clearly outlined, but the predicted hand and object masks and all masks show good localization to the held item of interest. The fourth row shows someone opening a cupboard. The all masks show both hands and portions of the held objects. The cupboard is well identified, though the prediction does not cover the entire cupboard door. The fifth row shows someone putting a lid on or pulling a lid off a container. The hand object masks are slightly muddled, however it is not clear that they are wrong as the persons hand wraps under the held object. The sixth row shows someone putting putting a plate on a drying rack. The plate's responsibility is well predicted, although a portion of the left side of the plate is not predicted, near where the other hand's responsibility would be if it were in contact with the plate. The seventh row shows someone opening the door of a washing machine. The predicted responsibility and hand object masks are well demarcated, though the object mask ends before the end of the door. The eighth row shows someone washing a pan in the sink. The object of interest is not too clear because of lighting, however the network still estimates the hands and portions of the objects successfully. The ninth row shows a transparent container grasped and successfully identified as a held object. The final row shows someone putting a baking sheet in (or taking it out) of an oven. Both hands have good coverage of the object, despite its large size.

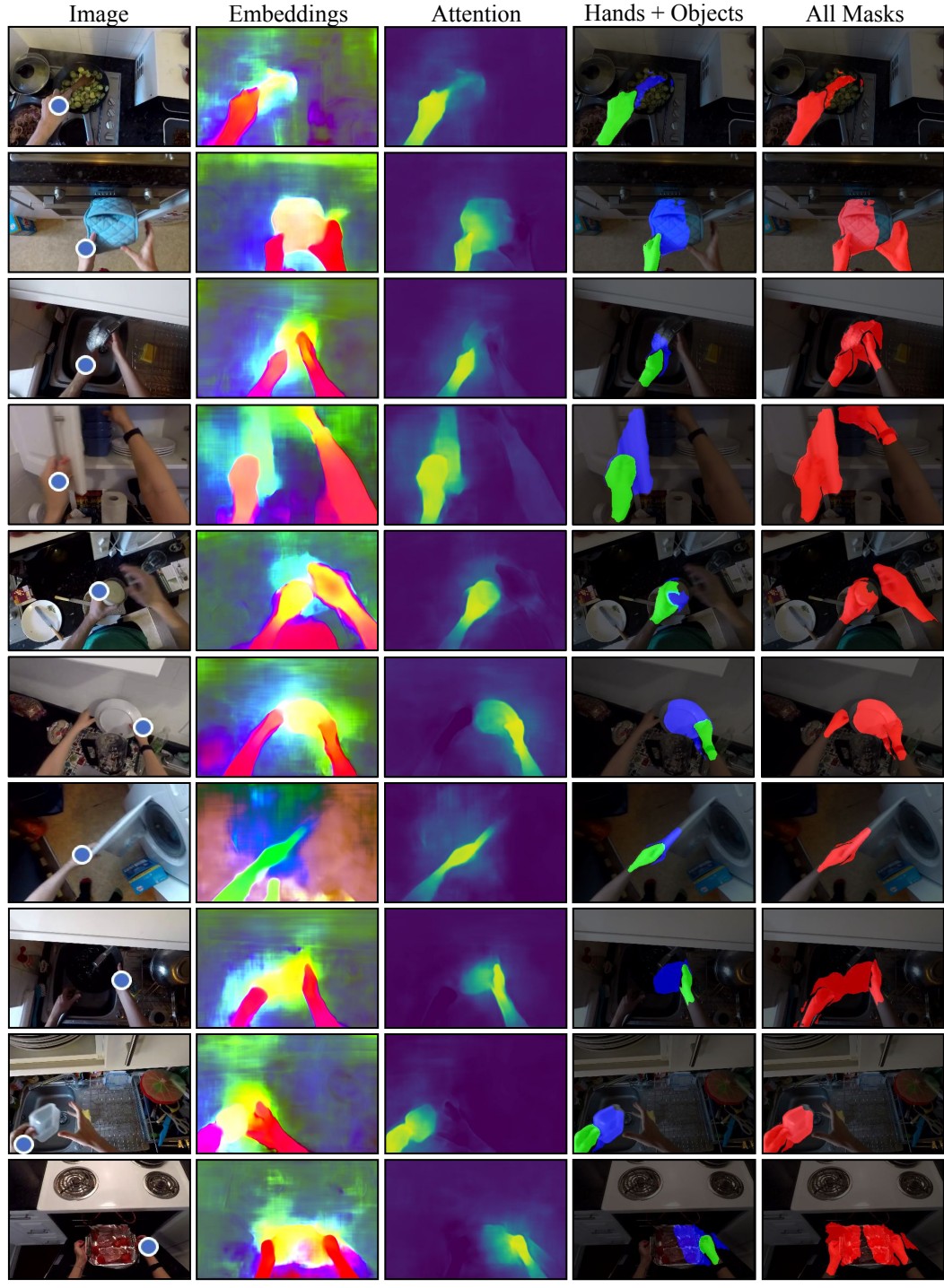

Figure 3: Additional selected examples of model output chosen from EPICK. The first column shows the image and input query point, the second column is a visualization of the model's embedding, the third column is the predicted responsibility, and the fourth and fifth columns show predicted hand/obj and all masks.

**Random results** Finally, we include 10 randomly sampled results, 5 from 100DOH and 5 from EPICK. The first column is the image, the second column the predicted responsibility attention map, the third and fourth column are PCA-ed and clustered outputs. The final column is the predicted pair mask for the hand associated with the dot in the first column image.

First we consider the 5 rows from 100DOH. The first row depicts the successful segmentation of the held milkshake, despite its varying colors. The straw is missed, which is typical of small objects from the dataset. The second row shows someone slicing food. The embedding is interesting, and from the embedding it seems that the held knife is similar in embedding to the object – a good result. The final mask does not properly cover the extent of the hand though. The third row depicts someone holding a sandwich. The responsibility prediction and mask are both successful segmentations, and the embeddings even capture the other people walking in the scene. The fourth row depicts someone using a blender, and the predicted responsibility map identifies their hand well, but struggles at extending this prediction to the full smoothie. The embeddings and predicted mask cover the hand excellent and object well, but also highlight the adjacent blender stand. This may not actually be a failure however, as it appears the back of the person's hand is in contact with this base. The fifth row depicts someone scooping ice cream or sorbet from a container. The predicted embedding, responsibility, and masks successfully identify the appropriate hand and the boundaries of the object.

Next we consider the 5 rows from EPICK. The first of these 5 rows depicts someone holding a plate and drying cloth. The embedding and responsibility look reasonable, and the all mask is excellent, despite the hand object mask misattributing part of the rag to the left hand. The next row depicts someone washing a plate in the sink. The predicted responsibility successfully captures the boundaries of the plate, and the embeddings improve upon this– grouping more of the plate together. The hand object mask is limited, only selecting half the plate, but as hypothesized in the original visualization, the all mask succesfully captures most of the other half of the plate. The 3rd of these 5 EPICK results rows shows someone washing something in a dark environment, and has reasonable but uncertain embeddings and predicted responsibility. However, the hand object prediction does a reasonable job of selecting part of the bowl. The next row is similar to this row. The final row shows someone scraping food off of a cutting board, and has a quite successful object embedding as depicted by the embeddings, responsibility, and masks. By looking at the all mask, we can further see that the other hand's predicted masks include the slim knife, though it also extends to the dish towel.

## F   Data Annotation

We used a crowdsourcing platform (thehive.ai) to label our data. The process of our annotation is as follows (1) annotate the hand mask for the hand with a red dot on it (2) annotate the object mask for the hand in red mask (drawn using the annotation from step1) if there exists an in-contact object.

## References

[1] Pavel Yakubovskiy. Segmentation models pytorch. `https://github.com/qubvel/segmentation_models.pytorch`, 2020.

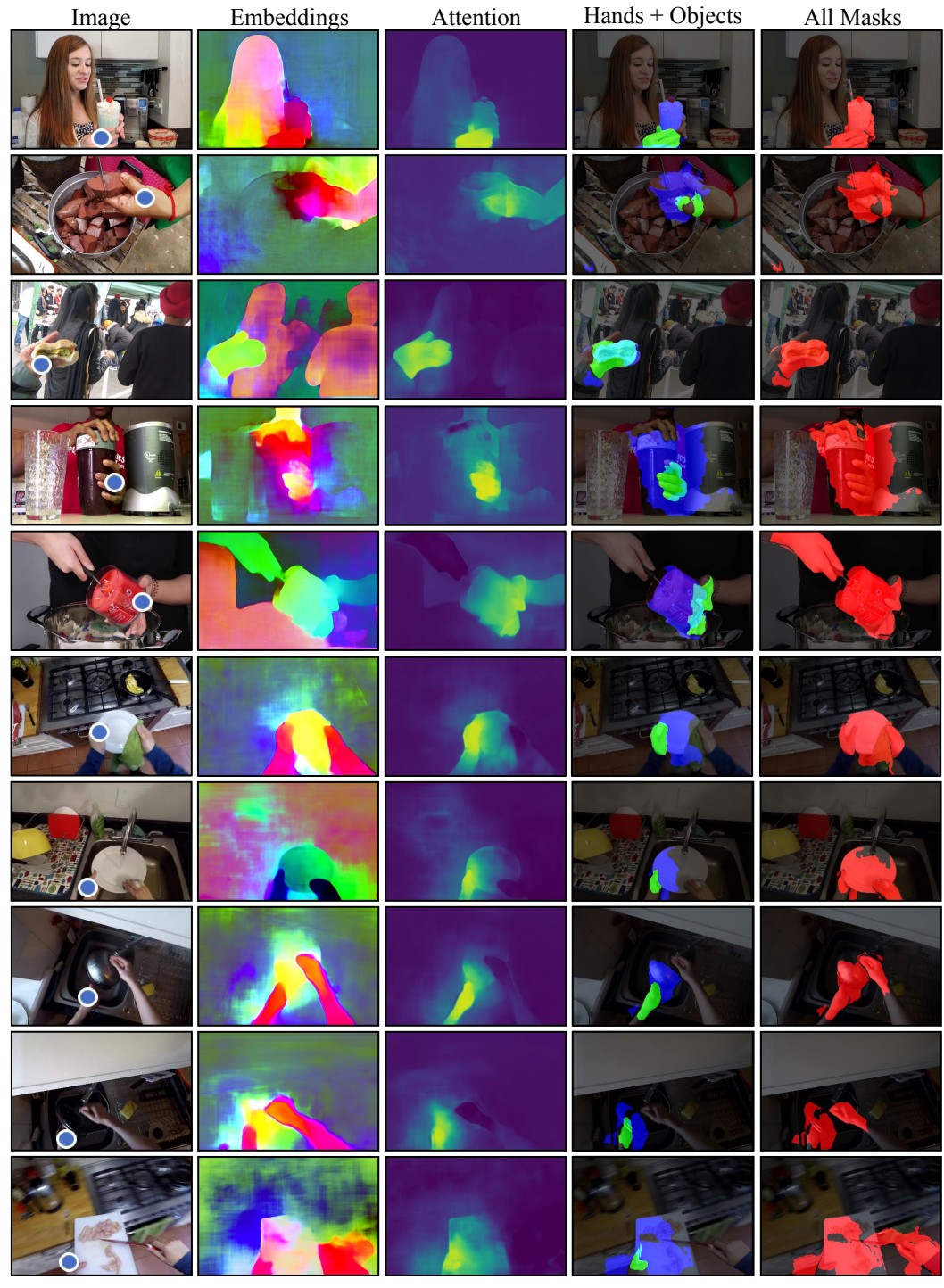

Figure 4: Random results from the test set of both 100DOH and EPICK. The first 5 rows represent results from 100DOH and the last 5 rows represent results from EPICK. To avoid resampling bias, we choose the same random samples selected in the original submission.

Segment one hand under the red dot in the image

B  *i*  U  S  AA▾  A  ☺  ▤▾  ▤▾  ▤▾  ▤  ▤  —  ▦  ▣  ▢

Given an image, only segment the hand that is under the red dot in the image.

The red dot indicates which hand to segment when especially there are multiple hands in the image. In the examples that there is more than one hand, you only need to segment the hand under the red dot. You do not need to segment other hands.

When the hand is occluded by the object, still segment out the whole hand area.

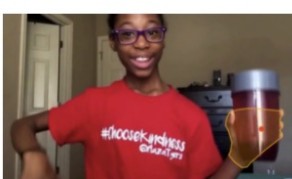

Figure 5: Data annotation step1: segment out hand under red dot in the image.

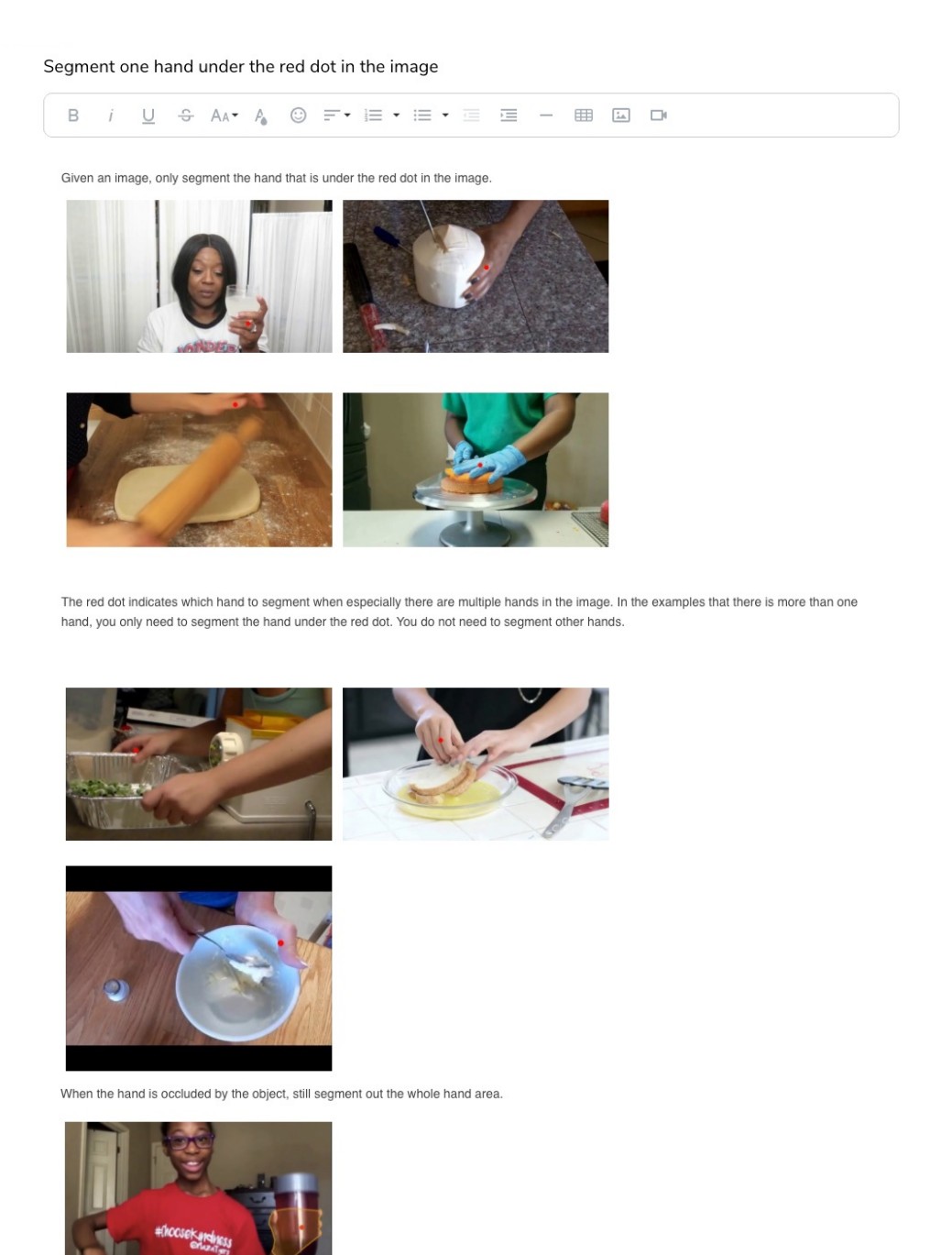

Figure 6: Data annotation step2: Segment the object in contact/moving together with the red hand.

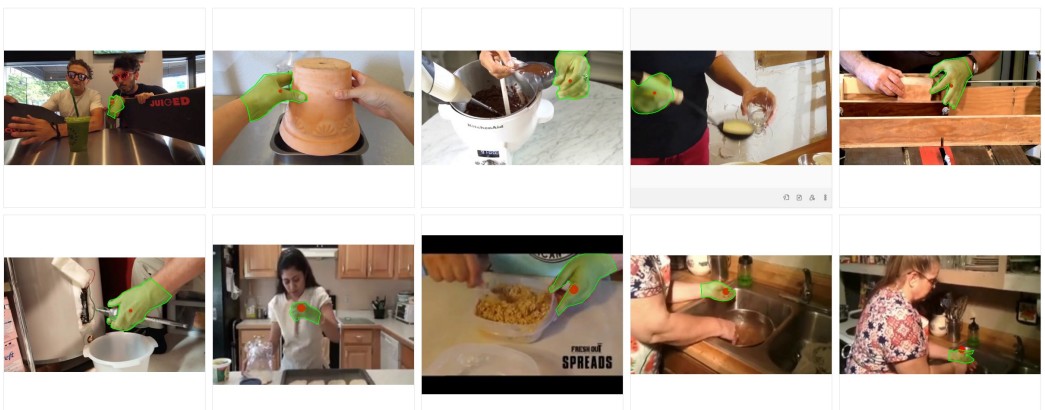

Figure 7: Data annotation step1 example. Given red dots, workers' annotations in green.

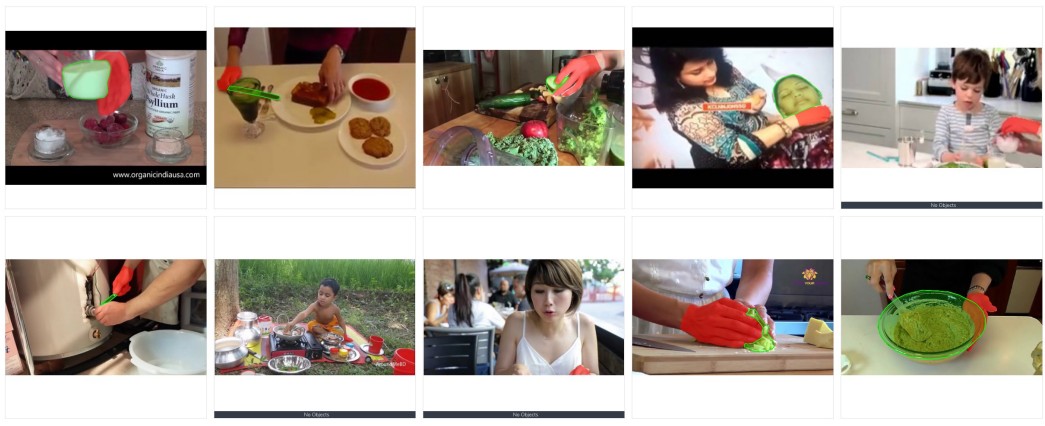

Figure 8: Data annotation step2 example. Given hands in red, workers' annotations in green.