# OpenReview forum: "COHESIV: Contrastive Object and Hand Embedding Segmentation In Video"
_NeurIPS.cc/2021/Conference — NeurIPS 2021 Poster_

### Official Review · Reviewer_obE8 · 2021-07-14

**Rating:** 7
**Confidence:** 3

**Summary:**

This paper addresses a novel problem of segmenting the object which is contacted by hand from a singe image. Since there are no annotations for this task, the authors present an approach to mine pseudo labels from videos based on optical flow. In particular, the authors assume that the contacted object has similar motion fields as the hand. The method works well for many cases except when objects are not moving or not moving in planar planes as also discussed in the paper. The experimental results demonstrate the effectiveness of the approach.

**Limitations And Societal Impact:**

Yes

**Main Review:**

The addressed problem is interesting and novel. In particular, the self-supervised learning style helps to generalize to a large number of contact objects which is expensive to annotate in supervised learning.

But the method is very simple. Using optical flow to find objects has been studied previously although not for contact object segmentation [1,2,3]. In addition, the method has limitations when objects are not moving as is also discussed in Figure 4. I believe handling static objects will make the paper stronger.

In general, the paper is well written and reads easily. Some minor issues:
(1) In line104, what does "responsibility minus people" mean?
(2) In training, is the number of hands N for different images fixed? Based on my understanding, the responsibility maps should have fixed shapes. But in practice, the number of objects can be very different from image to image.
(3) In line152, you mentioned to select data in which hands are contacting objects. How did you get this information?
(4) Reference [21] is incomplete.

One interesting baseline is missing. In inference, if we are given a pair of consecutive images, we can segment the contacted object using optical flow (as in training). How is the accuracy for that?

Ablation study with the contrastive loss should be provided.

[1]Romero, Javier, Matthew Loper, and Michael J. Black. "FlowCap: 2D human pose from optical flow." In German conference on pattern recognition, pp. 412-423. Springer, Cham, 2015.
[2]Fragkiadaki, Katerina, Han Hu, and Jianbo Shi. "Pose from flow and flow from pose." In Proceedings of the IEEE Conference on Computer Vision and Pattern Recognition, pp. 2059-2066. 2013.
[3]Wang, Chunyu, Yizhou Wang, and Alan L. Yuille. "An approach to pose-based action recognition." In Proceedings of the IEEE conference on computer vision and pattern recognition, pp. 915-922. 2013.

**Time Spent Reviewing:**

8

---

> ### Author Response · Authors · 2021-08-11
> **Author Response - Reviewer obE8**
>
> > The addressed problem is interesting and novel. In particular, the self-supervised learning style helps to generalize to a large number of contact objects which is expensive to annotate in supervised learning.
>
> We are glad that the reviewer found the problem interesting and novel and sees the value in learning about in-contact objects without supervision given the diverse classes of held-objects and expense of annotating segments for them. We address each piece of feedback below and illustrate some supporting experiments separately in a general comment meant for all reviewers.
>
> > But the method is very simple. Using optical flow to find objects has been studied previously although not for contact object segmentation [1,2,3].
>
> We thank the reviewer for their additional references and will incorporate a discussion of them if accepted. The closest is [3], which also uses flow as a cue for figure/ground separation (along with many others). Like some of this work, we are organizing only part of the scene (here the hands and objects). We see our responsibility approach as being similar, but using flow-based affinity with the hands to separate in-contact objects rather than the hands themselves. We see the simplicity of the model as a strength, but would note that there are a number of modeling details that are important to get right (e.g., we had substantial difficulty predicting flow as opposed to responsibility).
>
> > In addition, the method has limitations when objects are not moving as is also discussed in Figure 4. I believe handling static objects will make the paper stronger.
>
> Please see the general response for our comment.
>
> Addressing minor issues:
> > (1) L104, what does "responsibility minus people" mean?
>
> A thresholded responsibility mask (capturing joint motion of hands and held objects) where pixels that are part of the Ternaus [21] person segmentation have been subtracted and set to 0.
>
> > (2) In training, is the number of hands N for different images fixed? Based on my understanding, the responsibility maps should have fixed shapes. But in practice, the number of objects can be very different from image to image.
>
> This is a careful observation and correct insight about the structure of the data loader. To address this problem, we sample one hand from those in the image and predict the responsibility for that one hand by using the hand location as a query point.
>
> > (3) In line 152, you mentioned to select data in which hands are contacting objects. How did you get this information?
>
> The system provided in [38] is able to predict hand characteristics, including a 5-class hand contact state prediction. We treat the hand as being in contact if its prediction from [38] is in any class other than not-in-contact. We also include the constraints mentioned in Supplemental L55-62.
>
> > (4) Reference [21] is incomplete.
>
> We thank the reviewer for identifying our missing reference. The correct reference is below; we will include it in the updated reference section if accepted:
> Vladimir Iglovikov, et al. "TernausNet: U-Net with VGG11 Encoder Pre-Trained on ImageNet for Image Segmentation." (2018).
>
> > One interesting baseline is missing. In inference, if we are given a pair of consecutive images, we can segment the contacted object using optical flow (as in training). How is the accuracy for that?
>
> We thank the reviewer for this interesting baseline suggestion. We think the reviewer is referring to using the responsibilities produced for training as a prediction (since we already have a flow baseline that uses the raw flow magnitude).
>
> We have computed this. The responsibilities maps by themselves only permit evaluation on All and Pair settings. On All, for 100DOH, they obtain 44.5 mIoU (compared to proposed 46.4), and on Pair, they obtain 37.0 mIoU (compared to proposed 41.7). This performance validates our method, as we have demonstrated that our technique surpasses the capabilities of the pseudo labels it is trained against, an effect that was also seen, for instance in “Learning Features by Watching Objects Move” by Pathak et al., CVPR 2017. Note also that computing the responsibilities requires access to multiple frames on which to compute optical flow while COHESIV uses only one frame.
>
> | 100DOH | All | Pair |
> |--------------|:---:|:---:|
> | Proposed | 46.4 | 41.7 |
> | Thresholded Responsibility | 44.5 | 37.0 |
>
> > Ablation study with the contrastive loss should be provided.
>
> Table 2 row 2 shows an Attention-Only ablation of COHESIV which has no contrastive loss (although note that the Epic-Kitchens numbers for using both loses are out of date and trained on a different backbone; see the note at the start of the general response for a network trained with the same backbone). There is reduced performance. We finally thank the reviewer for any further ablation suggestions they may be able to provide, as thresholded responsibility was an informative baseline.

---

> > ### Comment · Reviewer_obE8 · 2021-08-18
> > **Final score 7**
> >
> > The authors addressed most of my concerns and I tend to change the score to 7.

---

### Official Review · Reviewer_BKyv · 2021-07-15

**Rating:** 6
**Confidence:** 2

**Summary:**

This paper tackles the problem of hand and object segmentation. At inference, the model takes a single image and the 2D location of the hand and outputs a hand-object segment. The key idea is to extract supervision from optical flow by contrastive learning.

**Limitations And Societal Impact:**

1. Although the method is tested on two large datasets (100DOH and Epic-Kitchens), the performance on Table 1 is not convincing. Only full comparison is available for Bbox [38] which the paper is not able to match the performance.

2. Manual labeling for both 100DOH and Epic-Kitchen datasets prevents readers from reproducing the results as they are not publicly available. Also, HO3D (Hampali et al., CVPR 2020) is another recent dataset which contains hand-object segmentation ground truth. I would suggest evaluation for readers to improve understanding the performance of the proposed method.

3. The model requires both a single image and a query point at the hand. I am interested in how this point is obtained and failure cases when the query point is far away from the hand. Otherwise, I found it counterintuitive if the model is limited by the query point.

**Main Review:**

1. Most hand and object interaction work requires strong supervision, this work however is able to derive supervision from optical flow in videos which is advantageous and extendable to in-the-wild datasets.

2. Both quantitative and ablation studies demonstrate promising results from their approach.

**Time Spent Reviewing:**

4

---

> ### Author Response · Authors · 2021-08-11
> **Author Response - Reviewer BKyv**
>
> We thank the reviewer for identifying our promising results and leaving a detailed review. We agree that hand and object interaction research is dominated by strongly-supervised methods and that the field needs further contributions using optical flow for supervision, as in our work. We address a few concerns raised by the reviewer below.
>
> > Only full comparison is available for Bbox [38]
>
> Below we include a full comparison for additional flow and saliency baselines. We did not compare with Flow and Saliency on the Pair/Hand/Object metrics originally because we felt it was an unfair comparison to baselines that are agnostic to the presence of hands. While our model can take a hand-query point and generate a segmentation, baselines have no such capability. However, we have computed these and report them:
>
> | 100DOH    	| All | Pair | Hands | Objects |  Epic-Kitchens | All | Pair | Hands | Objects|
> |----------	|:---: |:---: |:---:  |:---:	|------------- |:---:|:---:|:---:|:---:|
> | COHESIV  	| 46.4 | 41.7 | 51.3 | 30.8 | 	COHESIV 	| 44.6 | 43.4 | 60.4 | 20.8 |
> | Saliency 	| 25.2 | 20.1 | 8.6 | 17.0 | 	Saliency	| 21.6 | 15.9 | 6.0 | 11.7 |
> | Flow     	| 29.3 | 21.5 | 12.9 | 12.1 | 	Flow    	| 15.4 | 11.9 | 6.2 | 6.6 |
>
>
> Both saliencey and flow are generated for the entire image and then evaluated directly against all four targets (all/pair/hands/objects). While we further optimized the baselines for this experiment, we discovered and also fixed minor errors in the baselines.
>
> > Bbox [38] which the paper is not able to match the performance.
>
> We stress that COHESIV is using pseudo-labels that are automatically obtained rather than using manual and costly marking of 156K+ exhaustively annotated frames (100DOH + Epic-Kitchens + other egocentric data). We do not see strictly outperforming supervised techniques as a requirement for self-supervised learning techniques. Further, on Epic-Kitchens, the simplified model reported above nearly matches [38]’s performance on Pairs (43.4 vs 44.8) and surpasses it on Hands (60.4 vs 53.8).
>
> We have also done an experiment which sees whether we can *improve* on [38], as mentioned to all reviewers in the general comment. By combining our predicted segmentations with [38]’s bounding boxes, our modified approach obtains substantial performance gains on both 100DOH and Epic-Kitchens.
>
> Overall though, we do not think it should be a requirement that self-supervised methods outperform strongly-supervised methods. We think that obtaining quite close results to such a strongly supervised method with just pseudo-labels for training is a good start.
>
> > Manual labeling for both 100DOH and Epic-Kitchens datasets prevents readers from reproducing the results as they are not publicly available.
>
> Both source datasets (100DOH, Epic-Kitchens) are publicly available, and it is our full intention to release our own annotations with the paper. Since we are operating in a new problem setting, we had to obtain our own annotations that are not available yet. These annotations were expensive to obtain and we want the community to be able to use them. We apologize if any aspect of the paper gave a different impression.
>
> > HO3D (Hampali et al., CVPR 2020) is another recent dataset which contains hand-object segmentation ground truth.
>
> We thank the reviewer for this suggestion. We trained and evaluated our model on HO3D. We slightly modified the data generation process to account for the smaller dataset (by taking video clip samples with more overlap) and slower speed (by increasing the offsets between frames used for flow).  Our results obtain similar performance compared to 100DOH and Epic-Kitchens and are also relatively close to Bbox [38].
>
> | HO3D                          | All | Pair | Hands | Objects |
> | -------------|:----------:|:---------:|:----------:|:---------:|
> | COHESIV | 45.8 | 45.8 | 47.3 | 29.5 |
> | Bbox [38]  | 57.5 | 57.5 | 53.8 | 39.1 |
>
> We think that successful validation on an additional dataset considerably strengthens the paper and will include this experiment if accepted.
>
> > The model requires both a single image and a query point at the hand. I am interested in how this point is obtained and failure cases when the query point is far away from the hand.
>
> Currently, for simplicity, we use the average joint location from the detected hand. This setup ensures that our experiments focus on segmenting the hands and objects rather than localizing the hands themselves. That said, we note that hand detection works quite well -- [35] reports a mAP of ~90%.
>
> We thank the reviewer for their interesting question of what happens when the query point is far from the hand. This behavior is not defined in the training data, and so we have seen a variety of results when the query point is off the hands. For some points, it picks up on the nearest hand (or at least a nearby hand). This behavior may be because the Q feature maps are relatively smooth since ConvNets prefer smooth outputs. Also for some points, we have observed far away points to create responsibilities that match the background. But for both sorts of behaviors, some input and background query points produce nonsense.
>
> However, we can also easily build a hand-detector into the COHESIV model, enabling it to run entirely on its own. We trained a few convolutional layers (just like the map from Z -> Q), Conv(64->32, 3x3)--> ReLU-->  Conv(32->8, 3x3)--> ReLU--> Conv(8->1, 3x3), that map the latent space to a 1D heatmap for hand center location. We keep everything up to Z fixed. Following pose estimation techniques, this model is trained to minimize a regression loss (L2) to a Gaussian placed at the center of the box reported by [35]. We evaluated this with PCK (% Correct Keypoints), with the ground truth being the predicted hand location from [35] (which gets ~90%+ mAP). We report PCK below, using two PCK thresholds: 5% and 10% of the image diagonal.
>
> |                           | PCK@5% of diagonal | PCK@10% of diagonal |
> |------------------------|:-----------------------------:|:-------------------------------:|
> | 100DOH              | 55%                           | 73% |
> | Epic-Kitchens    | 89%                             |   95% |
>
> We note though that 100DOH has much higher variation in hand scale compared to Epic-Kitchens, and so it is difficult to absolutely threshold PCK.
>
> If we integrate these predicted keypoints into our system (using the nearest predicted keypoint to the ground-truth query point), we obtain the following results. We report these as (Predicted) and show them in comparison with the previous method (External).
>
> | Predicted vs. GT Hand Keypoint | All | Pair | Hands | Objects
> |----------------------------------|:--------:|:----:|:----:|:---:|
> | 100DOH (External)           | 46.4 | 41.7 |  51.5 | 30.8 |
> | 100DOH (Predicted)         | 41.9 | 38.3 | 46.2 | 29.3 |
> | Epic-Kitchens (External)   | 44.6 | 43.4 | 60.4 | 20.4 |
> | Epic-Kitchens (Predicted) | 40.7 | 38.9 | 56.5 | 18.9 |

---

> > ### Comment · Reviewer_BKyv · 2021-08-26
> > **Response to authors**
> >
> > Thank you for the detail response. The authors addressed all my major concerns and I am happy to aise my score to 6.

---

> > > ### Comment · Area_Chair_fiJq · 2021-08-29
> > > **Please update the score on your review**
> > >
> > > Hi Reviewer BKyv,
> > > If you are interested and willing to update your score, please do so now.
> > > Thanks,
> > > The AC

---

### Official Review · Reviewer_mYxT · 2021-07-17

**Rating:** 6
**Confidence:** 4

**Summary:**

This paper aims at segmenting hand and object regions given an input image, optical flow, and location of hand (for some pixels), using a contrastive approach and measurement of features called "Responsibility".

**Limitations And Societal Impact:**

Authors mention limitations and potential negative societal impact.

**Main Review:**

The paper is not well-written and the content do not follow naturally. It looks like a draft at this point. There are many ambiguities, grammatical errors, and lack of explanations throughout the paper (see details below). I read the paper twice to make sure I am not missing something and also checked the appendix. It is still not very clear to me many details of the paper.

**Technical / Clarity:**

- Regarding Eq. (1), which is measuring "Responsibility", what is the motivation and what is it computing? There is not much explanation on what is the goal of it. The first question is what is h_k? It is only at Eq. (2) that this becomes somewhat clear. Is h_k indeed proj(M_k)? if so, it should be clarified in Eq. (1). Now assuming proj(M_k) is indeed the answer, if M_k is the output of a model, the best predicted responsibility is to predicts the optical flow per-pixel. The question is then how this corresponds to a particular hand and why this is useful? In the denominator of Eq. (1) there are multiple hands for different h_k. How this equation differentiates different hands, if each hand output is predicting the optical flow?


- On a related note, in lines 128 and 133 M_k is defined as both homography and model. Which one is it? In line 129 the M_k is defined as optical flow plus current index location, while in Eq. (2) the same M_k is compared only to the optical flow (and not the sum of optical flow and index). These two definitions are not the same, so which one is correct? Also it is not clear how M_k and h_k related to the Z (common embedding), Q (hand features), and K (object features) given by the model.  In architecture section, this is not clarified. how M is obtained? Is it an output of the model or estimated differently?


- Authors mention in lines 158 to 160 that at test time given a query q, a score q^T \times K_{i,j} can be measured. What does this score between hand and object feature maps indicate and how it is used for hand region or object region segmentation? Q^T \times K should be mostly zero, if the two heads truly separates the activation for hand and object, otherwise if the multiplication is not-zero it indicates the model is not correctly separating them. The description at lines 176 to 179 also does not clarify it. A lot of such details are missing  in the paper.


- In Section 3.3 In "Attention/Direct Query Prediction Loss" section, a L1 loss is measured between Q^T \times K and the responsibility map. Again why this product should equal the responsibility? Assuming responsibility map is mostly showing the optical flow (given Eq. (1)), why this loss makes sense? again not much motivation is provided.


- In "Contrastive Loss" in Section 3.3, it is not clear how the mask of the object is measured? It is not clear how much annotation is required here. again many important details are missing.


- Authors mention, they use model [38] to get bounding box and hand joints, together with weak perspective values. However, it is not clear how and where these annotations are used. Do you use just some hand joint locations or entire segmentation? In both losses, how annotations given by different models or ground truth (if anything used) are leveraged?


- In Section 4.1 and 4.2 authors talk about attention-only and embedding-only, while these terms and what they mean are not defined in the text. While the authors mention their architecture has attention head and embedding, it is not clear for attention-only or embedding-only which loss is applied and how they are measured.


- Section Metric: it is not clear what is the difference between All and pair. It seems in both cases the union of hand and object is considered as positive.


- For K-means clustering, authors mention there are 3 clusters, but images show 4 colors (yellow, red, blue, and green). Why there is such a difference?


- In the appendix it is not clarified how responsibility maps are measured.  How the annotations or other info required for measuring the responsibility map are acquired? When using multiple frames authors mention "We then compute the necessary features between these +/- 3 frames and the center frame to produce a single responsibility map". how these "necessary features" are computed?



- In the appendix it is explained for detecting object parts, a hand query (provided as input) and a background query is used and then subtracted from the responsibility map. Since no background is provided as input, how this is computed? No visualised result of this process is shown. If this approach works better than using PCA or clustering (shown in images 2 and 3), why are these results not shown. On the other hand if PCA or clustering work better, why are the results in tables not computed that way?



- The usage of term responsibility throughout the whole paper is very unconventional. This term has not been used in similar literature. I recommend replacing it with something that better conveys what the method is actually doing.


**Technical:**

- The major drawback of the proposed approach is that it requires hands and objects to be moving and background to be almost stationary, which is due to usage of optical flow. If the hand and object are not moving, there is no optical-flow signal. Also, if the background is moving rapidly together with hand (e.g. when camera is moving) it can produce a high optical flow which interferes with the optical flow of hand and object, so detection of them becomes erroneous. This limits the application of the proposed approach.


- In the contrastive section, it seems the labels for hand and object parts are already provided using [21] and union of responsibility. If such annotation is available one can directly train on labels and no contrastive or semi-supervised training is needed. On a related note, in Table 2 better results are obtained with attention-only. Authors do not explain why this is happening in lines 292 to 295. Is it because you have enough annotations that embedding is not needed anymore?


- Are the test sets composed of new/unseen actions interacting with new objects? It is important to have unseen objects and actions in the test set, otherwise it is not clear if the model really generalises to new actions and unseen objects that exist in the real world.



- In Figures 3, 4, and 5 in appendix (also images 2 and 3 in the main paper), it is not clear what are the outputs of the model for each one of all, pair, object, and hand given the process described in D.1. In particular, the question is how the model separates hand, object, and background in these cases.


- The responsibility map shows the model can detect the hand and object regions together, but the separation of these two parts is not accurate. It seems the contrastive loss that should separate them is not helping much in separating the two.




**Grammatical/writing Errors:**

A lot of grammatical errors are seen in the paper. Such errors are also seen commonly in Appendix.
- line 34, 35 :  needs rephrasing, the phrase reads as if you are segmenting the motion, while you are segmenting the 3D object and not the motion.

- lines 56, 57: The first paragraph of related work is referring to itself and different parts of proposed work, while it is still not clear at this point what authors are particularly doing. Instead, authors need to talk about more general concepts in the literature and then relate them to their approach. The whole paragraph needs rephrasing.

- lines 69, 70: the example is not clear enough. The monitor is then used or unused in their case and how does it impact the results? Is it a problem of annotation, or the way the models are trained?

- line 78: weird positioning of "our work falls into this category. "

- line 82 : "Rather than learn" -> Rather than learning

- lines 118, 119: following phrase should be corrected "Formally, we the  responsibility as a temperature-softened softmax per-pixel"

- lines 162, 163: Three models are mentioned that predict embedding Z and it is not clear which one is actually used.

- line 164: "In one cases" -> In one case

- line 164: it is not clear how the phrase in parenthesis relates to the previous phrase.

- lines 168 and 169: the phrases in parentheses are not connected to the main sentence and it interferes with the flow of the text.

- line 184: "these each require " -> each of these require

- Section 3.3.The usage of the term "direct" for the loss is very uncommon. If the loss is supervised, the term supervised should be used.

- line 192: 'after' can be replaced by 'once'. It is more academic.

- The usage of punctuations and other grammatical symbols have been casual throughout the text. The dot is sometimes missed.:
   - line 227: before "We also generate 30K "
   -line 237: before " This data is sufficiently "

- lines 243 to 245: the phrase is erroneous and should be rephrased.

- line 280: "adjacent pairs of video in an image " I guess authors mean "adjacent frames of a video". such writing is very ambiguous and is seen often in the text.

- line 339 "with their hands can have of course be used for surveillance purposes. " -> remove "have"




**Time Spent Reviewing:**

12 hours

---

> ### Author Response · Authors · 2021-08-11
> **Author Response - Reviewer mYxT - Part 1**
>
> We thank the reviewer for the review. We have ordered the response in a way that we hope will be conducive to discussion and have put more minor questions towards the end. However, we believe we have answered all questions.
>
> *Writing quality*
>
> We don’t think it is accurate to say that the paper “looks like a draft at this point.” Both other reviewers understood the paper and asked detailed questions showing comprehension. Reviewer obE8 comments that it reads easily. That said, the paper of course contains some typos, and we thank the reviewer for spotting them.
>
> *What is a responsibility map, what is the motivation, and why are you using the term?*
>
> Responsibility is “how well that hand explains the pixel’s motion compared to other hands and the background” (L102-103). See also L5, L34-35, L43-44, L117-118. Given a motion model for each hand and for the background, one computes their error in predicting a pixel’s optical flow. Equation 1 (being a soft-max-like function) converts these errors into a set of “likelihoods” that sum to 1. The likelihoods are meant to represent the chance that each hand or background causes, or is responsible for, the observed optical flow at a pixel. If all things go well, the responsibility map for a hand should be all the pixels that the hand can cause to move. As we say at L102, the term comes when one fits a GMM with the EM algorithm. In GMM fitting, the responsibility represents the chance that a cluster is responsible for generating the data point and is computed with a similar contrastive-like equation.
>
> *How are responsibilities computed and used? Is $M_k$ the output of a [COHESIV] Model?*
>
> Computing responsibilities via Equation 1 requires a motion model that can be compared with a pixel’s optical flow. We fit such a model $M_k$ for a hand $h_k$ by fitting a homography on the optical flow on hand landmarks (L128). We specifically fit on the estimated flow at estimated Frankmocap [35] joints (L130-131). These hand joint locations are just (x,y) locations and in principle any way to obtain at least four points would work. The hand $h_k$ is just a convenience variable meant to contain the information needed about the hand (e.g., joint locations) to avoid explicitly writing out the indices over the joints or homography fitting.
>
> Once responsibilities are computed, they provide losses for the COHESIV model (L107-108). The training responsibilities and homographies produced by the COHESIV model are never used after training. Instead: at test time we aim to predict the non-human parts of this responsibility map (L104) on new images.
>
> Specific questions:
> - “$M_k$ is defined as both homography and model. Which one is it?” Homographies are a subset of the space of models. Homographies are parametric models and are referred to as such by many courses and books (e.g., search for “homography model fitting”), and the paper repeatedly refers to motion models.
> - “what is $h_k$?” $h_k$ is a convenience variable that encapsulates the information needed from the hand. We will rephrase so we index over distance functions instead to avoid introducing $h_k$.
> - “Is $h_k$ indeed proj($M_k$)?” $M_k$ is a 3x3 homography (L128). The operator proj converts homogeneous coordinates to ordinary ones (L136), or $[x,y,z]$ to $[x/z,y/z]$ . Thus proj($M_k$) is not defined.
> - “If $M_k$ is the output of a model, the best predicted responsibility is to predict the optical flow per-pixel.” $M_k$ is not the output of a model. L128-130 states that we fit the model to optical flow at the joints of the hand $h_k$. Responsibility is not optical flow. Optical flow has 2 components, but for N hands, there are N+1 responsibility maps for hands plus background that pixel-wise sum to 1 (L117). A N+1 channel image cannot represent optical flow in general, or optical flow for N hands.
> - “In architecture section ...  how M is obtained?” $M_k$ is only used temporarily when computing responsibilities in Section 3.1 and is fit to the flow at Frankmocap joints (L130). $M_k$ is fit using OpenCV’s homography fitting code, but any code would work. $M_k$ is never predicted by the COHESIV model; it is only fit to the flow data.
>
> We suspect that some of these questions may arise from less familiarity with homographies and geometric model fitting. We will add information to the paper to make it easier to understand  without this knowledge.
>
> *How does the prediction with Z, Q, K work and why should it produce responsibilities?*
>
> Most of the questions about why certain feature maps behave in certain ways are answered by the fact that the network is trained to minimize the losses. At training time, given a feature q on a hand, the network minimizes a L1 loss between $q^T K_{i,j}$ and the responsibility for that hand that is produced by Section 3.1 (L190-197).  At test time, one predicts responsibility for a hand by getting q and then by performing $q^T K_{i,j}$ for all pixels i,j. If the network is trained well, then $q^T K_{i,j}$ ought to be a good predictor of the responsibility.
>
> We hope this answers specific questions:
> - “a L1 loss is measured between $Q^T \times K$ and the responsibility map. Again why this product should equal the responsibility?” The network is training so that $q^T K_{i,j}$ equals the predicted responsibility. Small note: $Q^T$ is undefined, as is $Q^T K$.
> - "at test time given a query q, a score $q^T \times K_{i,j}$ can be measured. What does this score between hand and object feature maps indicate and how it is used for hand region or object region segmentation?” By training so $q^T K_{i,j}$ equals the responsibility on the training samples, then we hope that it will be the test responsibility at test time.
>
> *How much annotation is needed for the Contrastive Loss in Section 3.3?*
>
> No annotation is required (L210-214). The required information comes from [21] and the information needed to compute responsibilities. All are the outputs of automated systems.
>
> *The system requires hands and objects to be moving.*
>
> Please see common response on static objects.
>
> *The system requires the background to be almost stationary.*
>
> Our experiments show that the system can be trained on datasets with non-stationary cameras. We test on Epic-Kitchens, which is an egocentric dataset with moving cameras. The response to Reviewer BKyv also includes results on HO3D (another egocentric dataset). In both cases, the model is able to extract signals because with a reasonable frame rate, one can extract considerable signal from optical flow even with some background motion. We hypothesize that we get good performance because a pixel associates with a hand only if it is a better explanation of the relative errors; there is no absolute thresholding. Certainly there can be accidental hand/background synchronization, but we have not seen this.
>
> *In the contrastive section, it seems the labels for hand and object parts are already provided using [21] and union of responsibility. If such annotation is available one can directly train on labels and no contrastive or semi-supervised training is needed.*
>
> We respectfully disagree with this assessment. While purely supervised learning with targets makes sense in regimes with ground truth labels, it often does not make sense with pseudo labels and weakly-supervised settings. As an illustrative example of this, if our responsibility pseudo labels are evaluated against the test set, their mIoU is actually worse than our models mIoU.
>
> | 100DOH | All | Pair |
> |--------------|:---:|:---:|
> | COHESIV | 46.4 | 41.7 |
> | Thresholded Responsibility | 44.5 | 37.0 |
>
> Using pseudo labels as output leads to an mIoU of 44.5 on All. Pseudo labels are not perfectly accurate and often exclude portions of the held object. This steered us towards methods such as a contrastive loss, which encourages the network to recognize the regularity of objects when producing embeddings. Our model's results are better than directly evaluating against the thresholded responsibility pseudo labels. Even if this were not true, we stress that the architecture is also a contribution.
>
> *in Table 2 better results are obtained with attention-only.*
>
> Thank you for noticing this. Our results with a slightly adjusted architecture that can train both losses on Epic-Kitchens shows improvement over the attention model alone.
>
> *Are the test sets composed of new/unseen actions interacting with new objects?*
>
> We follow the splits of the datasets used, which in turn represent current best practices for validating recognition systems. 100DOH splits by person/uploader id, which is the standard that most recognition systems use. We use the Epic-Kitchens unseen participant split, which shows a totally new participant in a totally new environment. This split is even more challenging than is usually expected.
>
> *The responsibility map shows the model can detect the hand and object regions together, but the separation of these two parts is not accurate.*
>
> *If this approach works better than using PCA or clustering (shown in images 2 and 3), why are these results not shown. On the other hand if PCA or clustering work better, why are the results in tables not computed that way?*
>
> The responsibility map (e.g. the ground-truth ones shown in Figure 1 and predicted ones in Figures 2, 3) won’t separate the objects from the hand because it is all the pixels whose motion is explained by the hand. We therefore evaluate this quantitatively and directly in Table 1. Further, the Figures’ Clustering and PCA results suggest that there is good separation between objects and hands in feature space.
>
> We note though, that while PCA and clustering is a reasonable way of visualizing the feature spaces, the clusters themselves don’t have intrinsic meaning. Specifically, Cluster i of the clustered results does not consistently represent hands or objects. Thus, we cannot use these without some post-hoc assignment between clusters and ground-truth masks.

---

> > ### Author Response · Authors · 2021-08-11
> > **Author Response - Reviewer mYxT - Part 2**
> >
> > More detailed responses to questions:
> > - “In line 129, $M_k$ is defined as optical flow plus current index location”. $M_k$ is not defined as the optical flow. The equation at Line 129 gives a constraint that $M_k$ “should fit”. It’s on homogeneous coordinates so the equation is using homogeneous equivalence $\equiv$ ( $[x,y,z] \equiv [a,b,c]$ iff there is a $\lambda$ such that $[x,y,z] = \lambda [a,b,c]$). The precise form that is minimized during homography fitting is detailed and left out, but can be found in Hartley & Zisserman, for instance. We will add a brief description if accepted.
> > - In Equation 2, the output is compared only to the optical flow (as opposed to the flow plus a pixel). The reviewer is correct. Either O, the flow field, has to include the indices or [i,j] needs to be added. We apologize for this typo and will fix this.
> > - “It is not clear how $M_k$ and $h_k$ related to the Z given by the model.” The two are not directly related. The COHESIV model aims to predict the responsibility map for a given hand pixel. This responsibility map is computed at training time using the motion models $M_k$.
> > - “Authors mention, they use model [38] to get bounding box and hand joints, together with weak perspective values. However, it is not clear how and where these annotations are used. Do you use just some hand joint locations or entire segmentation?” We believe the reviewer is asking about L142-144. First, a small correction: [35] is used to get hand-joints; [35] in turn uses [38] for hand detection. Please see L130 “We found the simple approach of fitting $M_k$ to the estimated flow at Frankmocap [35] joints to be most effective.” Thus, we use just the joints.
> > - “In Section 4.1 and 4.2 authors talk about attention-only and embedding-only, while these terms and what they mean are not defined in the text. While the authors mention their architecture has attention head and embedding, it is not clear for attention-only or embedding-only which loss is applied and how they are measured.” The attention loss is referred to (L190) as “Attention/Direct Query Prediction Loss”. The embedding only loss is the (L198) contrastive loss, which is the loss which operates on the embeddings. We apologize for any confusion and will re-phrase the “Contrastive Loss” as “Embedding Loss”.
> > - Difference between all and pair: “All” includes a single mIoU for *all* hands and held-objects in an image. “Pair” is evaluated on a per-hand and held-object pair basis. `All’ does not require being able to distinguish between different hands and objects held by different hands. One can think of All as being similar to semantic segmentation and Pair as being similar to instance segmentation .
> > - K-means 4-vs-3 components: Thank you for catching this. We did set k=4. We will update the caption.
> > - “In the appendix it is not clarified how responsibility maps are measured.” Equations 1 and 2 explain this entirely: the responsibility of a hand for a pixel is proportional to the the residual error in explaining the pixel’s optical flow when one fits a homography to the optical flow at the hand’s joints (Equation 2 and surrounding); this error is combined with other hands and the background using a softmax-like model (Equation 1) to generate a final value.
> > - “How the annotations or other info required for measuring the responsibility map are acquired? … how these "necessary features" are computed?” Please see L142-145.
> > - “Since no background is provided as input, how this is computed?” We arbitrarily chose a corner of the embedding to be the background. We have since found also simplified object decoding schemes that do not require a background embedding; we will include the simplified process in the final paper.
> >
> > *Typos and writing issues.*
> >
> > We will fix all of the reported issues that are typos. We will re-examine and rework the parts that have been identified as potentially confusing.

---

> > > ### Comment · Reviewer_mYxT · 2021-08-20
> > > **Response to authors**
> > >
> > > Thank you for the detailed response. It clarified on many ambiguities. Adding a diagram of the model, with different components that explains the flow of information from input to output of the model and how the variables in the diagram relate to the ones used in the paper can help a lot, which I strongly recommend.
> > >
> > > Regarding the information in the paper, it is important to mention what each variable represents and how it is obtained. For example, it was not clear if M_k was obtained by just applying standard homography or if it was output of a network predicting these parameters.
> > >
> > > Due to the detailed response and clarification on all issues, I raise my score from 3 to 6. I think paper's materials needs some major re-organization/clarification to inject the information provided here into the text.
> > >
> > > Some other comments:
> > > - Reference [21] is not complete, please fix.
> > >
> > > - If I understood correctly, the query q point represents a 2D location, so how multiplication of q^T  * K_{i,j} (line 159) indicates the feature of interest and not picking just K at location {i,j}. If q is indeed a feature map, this needs to be clarified.

---

> > > > ### Author Response · Authors · 2021-08-31
> > > > **Author Response - Reviewer mYxT - Part 3**
> > > >
> > > > Thank you for engaging in the discussion, updating your review and for your further helpful and thoughtful feedback.
> > > >
> > > > We apologize for the incomplete reference. We will be sure to correct it in the updated version. The correct reference for [21] is below: Vladimir Iglovikov, et al. "TernausNet: U-Net with VGG11 Encoder Pre-Trained on ImageNet for Image Segmentation." (2018).
> > > >
> > > > We are committed to ensuring that the paper is as readable as possible, and will be editing paper and supplemental to maximize readability. We think R1’s suggestion of a diagram that includes more details is a great idea and we are sorry that the figure in the supplemental does not cover all the details. To demonstrate our commitment towards clarifying the paper, we have put an anonymous preview of an updated diagram that covers both the responsibility generation and COHESIV architecture. It contains both the inputs and outputs and the relationships between the variables. You can see it at this link: https://i.ibb.co/gyd3G6j/cohesiv-diagram.jpg
> > > > We will incorporate this in the supplemental, or (if we can make space by some editing in the discussion), the main body of the paper.
> > > >
> > > > We apologize for the confusion about the query point location (say, for instance, p=(x, y)) and query vector q. To be clear, the input query point location p=(x, y)  is the 2D hand location in the RGB image. Then, we take the feature under location p on the Q feature map as the query vector q to calculate the score $q^{T}K_{i,j}$. We also clarify this in the diagram provided above.

---

### Author Response · Authors · 2021-08-11
**General Comment**

We thank the reviewers for their feedback on the current paper. We are glad that reviewers thought our problem and approach are promising, novel, important, and finally advantageous and extendable given the minimal supervision requirements.

Reviewer mYxT had a number of confusions about the paper, which we address in specific comments. Reviewer BKyv pointed out a few experimental limitations, which we address with additional experiments. Finally, reviewer obE8 was overall positive, but had a few questions.  We will address a few common concerns having to do with static objects, common experimental questions, and experiments that are likely of interest to all reviewers.

Before we begin, we would like to report that we have since found that a few architectural simplifications (dropping CoordConv, plus embedding normalization and batchnorm) enables the backbone (se-resnext50-4d) to train consistently on all datasets. This improved system obtains mostly similar results on 100DOH, and reasonable improvements on Epic-Kitchens.

|                        | All | Pair | Hands | Objects |
|--------------------|:------:|:------:|:------:|:-------:|
|100DOH          | 46.4 | 41.7 | 51.5 | 30.8 |
| Epic-Kitchens | 44.6 | 43.4 | 60.4 | 20.8 |

One reviewer (BKyv) was concerned about our performance relative to the supervised [38]. We address this in more detail, but one experiment that may be of broader interest is a combination of our system’s predicted segmentations and the [38] system. Specifically, we demonstrate that predicted segmentations are complementary to the [38] system, and the combination of methods achieves superior performance to [38] alone.

| 100DOH | All | Epic-Kitchens | All |
| --- | :---: | --- | :---: |
| Bbox [38]			| 56.9 | Bbox [38]			| 54.3 |
| COHESIV			| 46.4 | COHESIV			| 44.6 |
| COHESIV + Bbox [38]	| 61.5 | COHESIV + Bbox [38] 	| 56.6 |

Reviewer BKyv was also interested in results on HO3D. This may be of interest to other reviewers and since it is an egocentric dataset, it helps further address Reviewer mYxT’s question about whether our system requires a static background.

| HO3D                          | All | Pair | Hands | Objects |
| -------------|:----------:|:---------:|:----------:|:---------:|
| COHESIV | 45.8 | 45.8 | 47.3 | 29.5 |
| Bbox [38]  | 57.5 | 57.5 | 53.8 | 39.1 |

A few reviewers asked about static objects (e.g. shown in Figure 4). We think tackling (almost) always static objects (for instance shown in Figure 4) is an open question to be solved by the community over a decade rather than in one paper. Generic object individuation without resorting to extensive supervision like BSDS has been a goal of the community for decades. In our view, getting reasonable performance on held-objects by assuming knowledge of people and flow is a strong first start. It may be possible to use a video to transfer segmentations from in-motion to static objects. However, given the considerable challenge of the problem, especially using data like 100DOH and Epic-Kitchens, we see this as a research direction rather than a project. As it happens, the example of Figure 4 is particularly challenging since the object itself is a full-sized coffee machine, possibly requiring two people to move it.

---

### Decision · Program_Chairs · 2021-09-27

**Decision:**

Accept (Poster)

**Comment:**

The reviewer ratings of this paper were significantly lower prior to the author response.
Reviewers were concerned about the quality of the writing.
Additional experiments, clarifications and even some simplifications of the proposed method were provided during the author response.
As a result of these actions all reviewers increased their scores after reading the author response.
After accounting for the author response and discussions with reviewers, all reviewers recommend accepting this paper.
It will be important for the authors to address the points and issues raised during the review and author discussion in the final version of this paper.

The AC recommends accepting this paper.